# Speaking without vocal folds using a machine-learning-assisted wearable sensing-actuation system

Ziyuan Che[1,2], Xiao Wan[1,2], Jing Xu [1], Chrystal Duan[1], Tianqi Zheng[1] & Jun Chen [1] ✉

Voice disorders resulting from various pathological vocal fold conditions or postoperative recovery of laryngeal cancer surgeries, are common causes of dysphonia. Here, we present a self-powered wearable sensing-actuation system based on soft magnetoelasticity that enables assisted speaking without relying on the vocal folds. It holds a lightweighted mass of approximately 7.2 g, skin-alike modulus of $7.83 \times 10^5$ Pa, stability against skin perspiration, and a maximum stretchability of 164%. The wearable sensing component can effectively capture extrinsic laryngeal muscle movement and convert them into high-fidelity and analyzable electrical signals, which can be translated into speech signals with the assistance of machine learning algorithms with an accuracy of 94.68%. Then, with the wearable actuation component, the speech could be expressed as voice signals while circumventing vocal fold vibration. We expect this approach could facilitate the restoration of normal voice function and significantly enhance the quality of life for patients with dysfunctional vocal folds.

Voice, as the carrier wave of speech signals in human communication, is a vital component that underpins social interaction and artistic propagation. It serves as the melody of our speech and infuses our daily-articulated thoughts with expression, emotion, intent, and mood. Due to its significance in fostering integration between individuals and their communities, disorders with vocal folds, the essential voice-generating organ of humans, have a pronounced and objectionable impact. Voice disorders are generally defined as the condition where the malfunction of the laryngeal mechanism causes a person's voice quality, pitch, and loudness to differ from those of a population with similar demographic characteristics[1–4]. Under clinical circumstances, voice disorders result from assorted pathological conditions, including vocal fold polyps[5], keratosis[6,7], vocal fold paralysis[8,9], vocal fold nodules[10,11], and adductor spasmodic dysphonia[12,13]. Moreover, artificial medical interventions like laryngeal cancer surgeries may also cause temporary dysphonia due to the loss of control of vocal fold-related muscles[14–16]. Specifically, 29.9% of the general population had

at least one voice disorder during their lifetime, 7% are currently undergoing voice problems, and 7.2% of employed participants reported missing work days due to voice disorder[17]. Despite the prevalence of voice disorders across all ages and demographic groups and the effectiveness of therapeutic approaches such as voice therapy and surgical interventions, the recovery time can be burdensome[18–23]. Patients often require a recovery phase of three months to a year, with a postoperative period of absolute voice rest[10,24–30]. Existing solutions, such as handheld electrolarynx devices or alternatives like the "talk box" device and tracheoesophageal puncture procedures, can be inconvenient, uncomfortable, or invasive[31–34]. Therefore, there is a pressing need to develop a wearable, noninvasive medical device capable of assisting patients in communicating during the pre- and post-treatment recovery of voice disorders.

Existing research on medical devices using flexible loudspeakers and wearable throat sensors made from materials like polyvinylidene fluoride (PVDF)[35–38], gold nanowires[39], or graphene[40,41], has shown

[1]Department of Bioengineering, University of California, Los Angeles, Los Angeles, CA 90095, USA. [2]These authors contributed equally: Ziyuan Che, Xiao Wan. ✉e-mail: jun.chen@ucla.edu

potential for aiding communication during recovery from vocal fold disorders. PVDF emerges as a pristine thermoplastic fluoropolymer, notable for its exceptional non-reactivity[42]. A distinguishing feature of PVDF is its piezoelectric property, adeptly converting mechanical oscillations into precise voltage signals[43,44]. While this piezoelectric property offers certain advantages, the material selection for piezoelectric sensors remains limited, often constraining the design and functionality of devices tailored for specific applications. Also, even though piezoelectric materials present actuation abilities, the driving voltage would induce safety concerns for wearable bioelectronics. In parallel, gold nanowires and graphene have gained recognition for their superior conductivity and inherent flexibility. These characteristics make them ideal candidates for crafting resistive sensors, which can swiftly measure the resistance changes in response to mechanical stresses. However, these resistive sensors, including those made from gold nanowires, typically require an external power source for sensing, adding to the complexity and potential bulkiness of the wearable system. Furthermore, despite their impressive attributes, the inherent non-stretchable nature of these materials poses a significant limitation. They predominantly detect vertical throat movements, often neglecting the parallel deformation that occurs during phonation, which involves a complex interplay of various laryngeal muscle groups. These muscles, including extrinsic[45–49] and platysma[50,51] muscles, contribute to throat movement during phonation and are particularly important for patients with voice disorders who cannot use their vocal folds[45,46,52]. Additionally, non-stretchable materials can affect comfort and adhesiveness. And other issues of those materials such as lack of water (perspiration) resistance and temperature rise, can lead to operational problems.

Here, we present a wearable and self-powered sensing-actuation system based on soft magnetoelasticity as a fundamentally new platform technology for assisted speaking without vocal folds. The system allows patients to articulate sentences solely through muscle movements associated with regular speech or lip-synching. The sensing component of the system detects the extrinsic laryngeal muscle movements without the vibration of vocal folds. To enhance the sensitivity, we designed the kirigami structure of the sensor with enlarged unit horizontal and vertical deformation, thus generating high-quality electrical signals for downstream processing. These electrical signals are fed to a pre-trained machine-learning model that converts throat movement into voice signals. The system exhibits high sensitivity, a quick response time of 40 ms, a lightweight mass of 7.2 g, and possesses a skin-alike modulus of $7.83 \times 10^5$ Pa, ensuring accuracy and wearing comfort. Furthermore, a stretchability of 164% for horizontal deformation detection enhances adhesive attachment of the device to the throat, contributing to precise movement detection, tackling the crucial issue of capturing omnidirectional mechanical deformation. The magnetoelastic property of the material enables both sensing and actuation in one soft and stretchable system. The system is intrinsically waterproof since the magnetic field is not attenuated by water, ensuring durability and functionality even in the presence of heavy perspiration. Towards practical application, we have demonstrated that the wearable sensing-actuation system is able to perform daily language transmissions and clear output of voice with an accuracy of 94.68%. These results establish the foundation for a potential solution to voice disorders by facilitating voice usage in patients with voice disorders during their recovery period, offering opportunities to enhance their overall quality of life.

## Results
### Design of the wearable sensing-actuation system
A thin, flexible, and adhesive wearable sensing-actuation system was attached to the throat surface, as shown in Fig. 1a, for speaking without vocal folds. This system comprises two symmetrical components: a sensing component (located at the bottom part of the device)

converting the biomechanical muscle activities into high-fidelity electrical signals and an actuation component using the electrical signals to produce sound (located at the upper part of the device), as shown in Fig. 1b. Both components consist of a polydimethylsiloxane (PDMS) layer (~200 μm thick) and a magnetic induction (MI) layer made of serpentine copper coil (with 20 turns and a diameter of ~67 μm). The serpentine configuration of the coil ensures the flexibility of the device while maintaining its performance, as discussed in Supplementary Note 1. The symmetrical design of the device enhances its user-friendliness. The middle layer of the device is the shared magneto-mechanical coupling (MC) layer, made of magnetoelastic materials consisting of mixed PDMS and micromagnets. The MC layer, with a thickness of approximately 1 mm, is fabricated with a kirigami structure to enhance the device's sensitivity and stretchability (see Fig. S1). The entire system is small, thin (~1.35 cm³, with a width and length of ~30 mm and a thickness of ~1.5 mm), and lightweight (~7.2268 g) (see Fig. S2 and Supplementary Table S1).

Multidirectional movement of laryngeal muscles sets the significance of capturing laryngeal muscle movement signals in a three-dimensional manner. Moreover, the learning process of phonation may be heterogeneous across populations: different people may adopt a variety of muscle patterns to achieve identical vocal movements[45,53]. Such complexity of muscle movement requires the device to be able to capture the deformation of muscles not horizontally or vertically alone, but rather in a three-dimensional way. Figure 1c illustrates the movement of the muscle fiber during two stages, i.e., expansion and contraction. During the expansion phase, the muscle relaxes and elongates in the $x$- and $y$-axis. On the other hand, during the contraction phase, the muscle shortens in the $x$- and $y$-axis while thickening in the $z$-axis through the increase in muscle fiber bundle diameters. Figure 1d, e demonstrates the device's response in the $x$-, $y$-axis, and $z$-axis, respectively. During the expansion phase, the kirigami-structured device expands in surface area with slight deformation in the $z$-axis. Conversely, during the contraction phase, the device opposes deformation in the $x$- and $y$-axis and undergoes deformation in the $z$-axis. Thus, the device captures the muscle movement across all three dimensions by measuring the corresponding deformation, which generates the change of magnetic flux density followed by the induction of an electrical signal in the MI layer. Supplementary Note 2 further demonstrates the response of the device to the omnidirectional laryngeal movements and how the kirigami structure ensures the sensing performance.

The key defining characteristic of this system (MC layer) is based on the magnetoelastic effect, which refers to a change in the magnetic flux density of a ferromagnetic material in response to an externally applied mechanical stress, which was discovered in the mid-19th century[54]. It has been observed in rigid metals and metal alloys such as $Fe_{1-x}Co_x$[54], $Tb_xDy_{1-x}Fe_2$ (Terfenol-D)[55], and $Ga_xFe_{1-x}$ (Galfenol)[56]. Historically, these materials received limited attention within the bioelectronics domain for several reasons: the magnetization variation of magnetic alloys within biomechanical stress ranges is limited; the necessity for an external magnetic field introduces structural intricacies; and a significant mechanical modulus mismatch exists between magnetic alloys and human tissue, differing by six orders of magnitude. However, a breakthrough occurred in 2021 when the pronounced magnetoelastic effect was observed in a soft matter system[57]. This system exhibited a peak magnetomechanical coupling factor of $7.17 \times 10^{-8}$ T Pa$^{-1}$, representing an enhancement up to fourfold compared to traditional rigid metal alloys, underscoring its potential in soft bioelectronics. Functionally, the MC layer converts the mechanical movement of extrinsic laryngeal muscle into magnetic field variation, and the copper coils transfer the magnetic change into electrical signals based on electromagnetic induction, operating in a self-powered manner. While additional power management circuits are essential for processing and filtering the signals, the initial sensing phase is

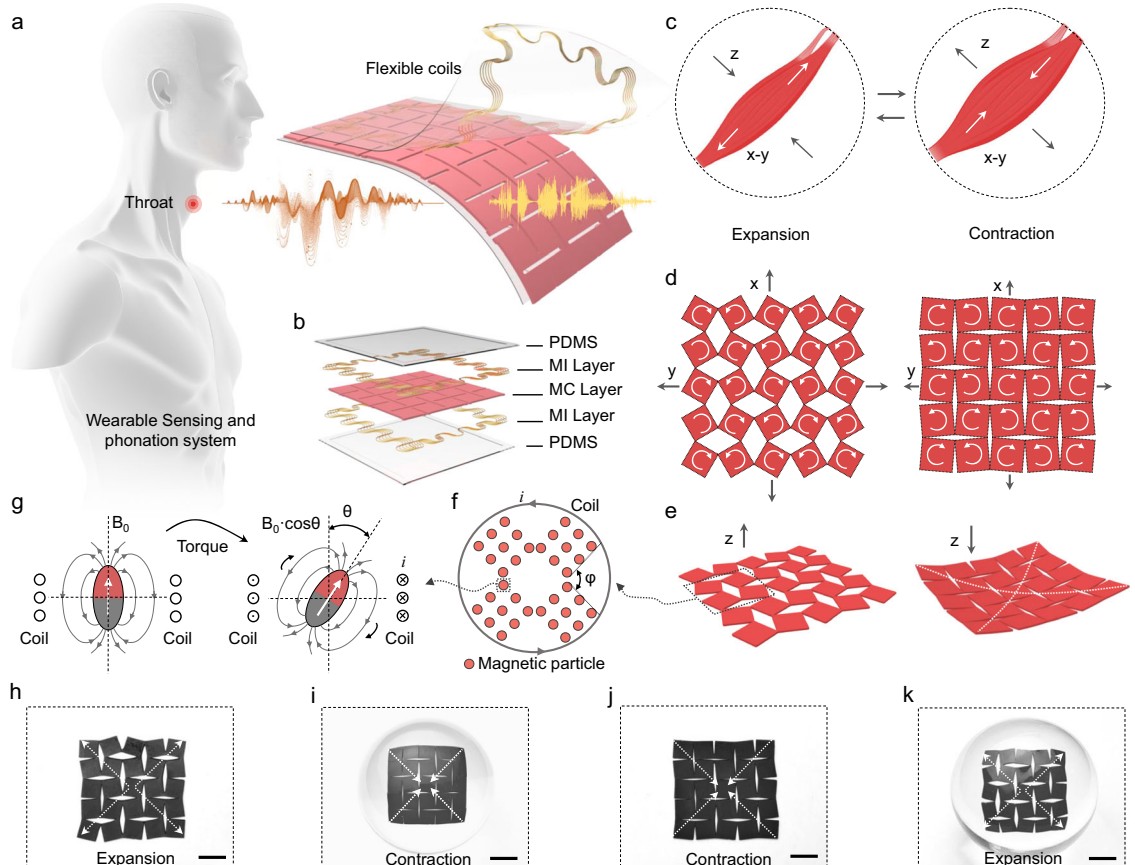

**Fig. 1 | Design of the wearable sensing-actuation system. a** Illustration of the wearable sensing and phonation system attached to the throat. **b** Explosion diagram exhibiting each layer of the device design. **c** Two modes of muscle movement, expansion induces the elongation in the *x*- and *y*-axis, while contraction induces the elongation in the *z*-axis. Kirigami-structured device response to muscle movement patterns in the x, y (**d**), and z direction (**e**): expansion results in *x*- and *y*-axis expansion and less deformation in the *z*-axis, contraction results in less deformation in x and y direction and expansion of the *z*-axis. **f** Detailed illustration of the magnetic field change caused by magnetic particles. For one part, the angle change between each single unit of the kirigami structure is represented by $\varphi$. For the other part, the magnetic particle itself undergoes torque caused by the deformation applied onto the polymer (**g**), thus, generating a change of magnetic flux and, subsequently, current in the coil. The photo of the device in muscle expansion state is shown in **h** (*x*-, *y*-axis), **i** (*z*-axis), and in muscle contraction state is shown in **j** (*x*-, *y*-axis), **k** (*z*-axis). Scale bars, 1 cm.

autonomous and does not rely on an external power supply. After recognition through the machine learning model, the voice signal is output through the actuation system (Fig. 1a).

The signal conversion through the giant magnetoelastic effect in soft elastomers can be explained at both the micro and atomic scales. At the microscale, compressive stress applied to the soft polymer composite causes a corresponding shape deformation, leading to magnetic particle-particle interactions (MPPI), including changes in the distance and orientation of the inter-particle connections. The horizontal rotation of each subunit in the kirigami structure (Fig. 1d) and vertical bending deformation (Fig. 1e) create a micro change of magnetic density. In detail, as shown in Fig. 1f, in a subunit of the kirigami structure, deformation-induced angle shift $\varphi$ generates a concentration of stress and MPPI in between each single unit of the kirigami structure. At the atomic scale, mechanical stress also induces magnetic dipole-dipole interactions (MDDI), which results in the rotation and movement of magnetic domains within the particles. As shown in Fig. 1g, a torque was made on each magnetic nanoparticle, and the shift of angle $\theta$ generates the change in magnetic flux density. The photo of the device design is presented in Fig. 1h, i as the *x*-, *y*-axis, and *z*-axis response in the expansion phase; and in Fig. 1j, k as in the contraction phase. Fig. 1h, j describes the expansion and contraction in the x-y plane, and Fig. 1i, j describes the corresponding *z*-axis contraction and expansion. Such structural design also displays a series of

appealing features, including high current generation, low inner impedance, and intrinsic waterproofness, which will be presented in the following sections.

## Standard characterization

Our present work compares previous approaches based on PVDF and graphene for flexible voice monitoring and emitting, as shown in Fig. 2a and Supplementary Table S3[35–37,58–62]. The device developed in this work has a similar acoustic performance, with a frequency range covering the entire human hearing range. However, it has a much lower driving voltage (1.95 V) and a Young's modulus of $7.83 \times 10^5$ Pa. As shown in Fig. S3, it exhibits the stress-strain curve and testing photo of the material with and without the kirigami structure, which lowered Young's modulus from $2.59 \times 10^7$ Pa to $7.83 \times 10^5$ Pa. This result ensures a higher comfort level while wearing as the modulus of the device is very close to that of the human skin. Notably, the device we developed has two unique features of stretchability and water resistance, which ensure the detection of horizontal movements, wearing comfort and resistance to respiration. Additionally, the device does not have the issue of temperature rising during use, preventing unexpected low-temperature scalding of users. Subsequently, several standard tests establish the sensing features of the device and its efficacy in outputting voice signals. To enhance the stretchability of the device, a kirigami structure was fabricated onto the MC layer of the device. The

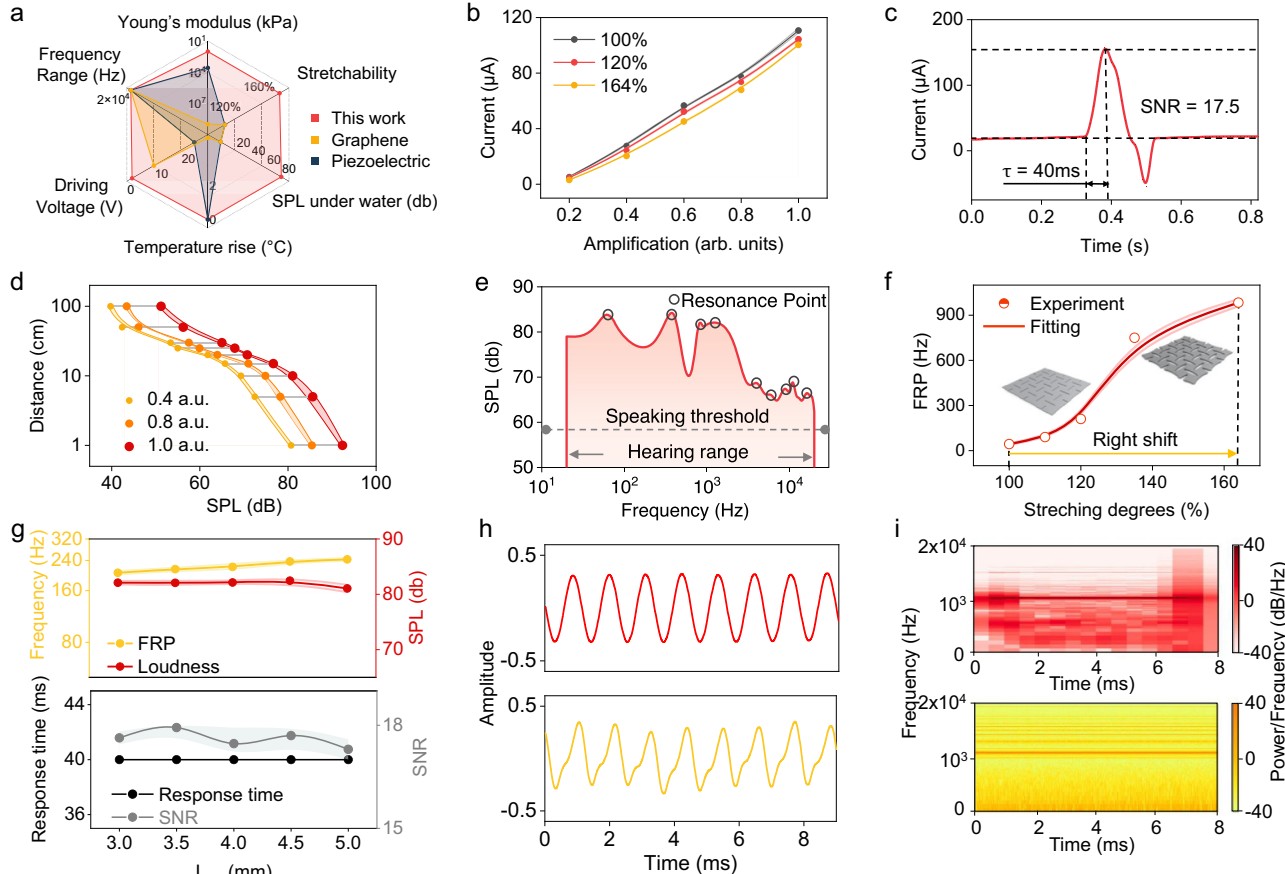

**Fig. 2 | Performance characterization of the wearable sensing-actuation system. a** Performance comparison of different flexible throat sensors in terms of Young's modulus, stretchability, underwater sound pressure level, temperature rise, driving voltage, and working frequency range. **b** Pressure–sensitivity response of the device at varied degrees of stretching under different amplification levels. (arb. units) referring to arbitrary units. **c** Response time and signal-to-noise ratio of the device. **d** Variation of sound pressure level with distance from the device at different amplification levels. **e** Sound pressure level of the device with resonance point highlighted in human hearing frequency range compared to SPL normal human speaking threshold. **f** The right shift of the first resonance point towards high frequency with regards to increasing strains. The test of performance is repeated 3 times at each condition. Data are presented as mean values ± SD. **g** Relationship between Kirigami structure parameters and actuating (first resonance point and sound pressure level)/sensing properties (response time and signal-to-noise ratio). The test of performance is repeated three times at each condition. Data are presented as mean values ± SD. Waveform (**h**) and spectrum (**i**) comparison of commercial loudspeaker (Red) and the device (Yellow) sound output at 900 Hz and maximum strain (164%).

unit design of the structure is shown in Fig. S4, and the stretchability with regards to the parameters of the kirigami unit design is exhibited in Fig. S5. Such an approach not only enhances the stretchability of the device to a maximum of 164% with Young's modulus at the level of 100 kPa but also realizes isotropy. Furthermore, the structure enlarges the horizontal deformation of the device under unit pressure, generating a higher current output and enhanced detectable signals of extrinsic muscle contraction and relaxation, as shown in Figs. S6 S7. The change in sensitivity brought by the structure on the vertical axis was also tested, and an elevation can be observed, as shown in Figs. S8 S9. Moreover, isotropy prevents the device from being disturbed by random and uneven body movements in use. Thus, there are no requirements on wearing orientation which elevates user-friendliness as revealed in Figs. S10 S11.

The stretchable structure of the device was leveraged to examine its sensitivity with respect to deformation degrees, as depicted in Fig. 2b. The sensitivity curve demonstrated consistency under varying strains, with a minor change observed under maximum strain (164%). This change could be attributed to the reduction in the MC layer's thickness due to deformation, which in turn decreases the magnetic flux density under the same pressure level, resulting in lower current generation. The device's response curve under different frequencies

and forces of the shaker was tested, as shown in Fig. S12. We have also validated that the electric output of the device is not due to the triboelectricity in Supplementary Note 3[63]. The device's inherent flexibility and stretchability facilitate tight adherence to the throat, yielding a high signal-to-noise ratio (SNR) and swift response time (Fig. 2c). In addition to the kirigami structure design parameters, other factors influencing the device's sensitivity, response time, and SNR were also evaluated. Fig. S13 illustrates that an increase in coil turns results in longer response times and lower SNR due to the increased total thickness of the copper coils. This thickness impedes the membrane's deformation during vibrations, leading to longer response times and lower signal quality. We have further investigated the increase of thickness with the coil turn ratios in Supplementary Table S2. As the number of coil turns escalates, there's a direct correlation with the likelihood of copper wires stacking. Consequently, a significant number of samples exhibit thicknesses approximating 2 or 3 layers of copper (134 μm and 201 μm, respectively). This stacking effect amplifies the average coil thickness as the number of turns increases. However, this augmentation isn't strictly linear. For instance, the propensity for overlapping is less pronounced for turn ratios of 20 and 40. In contrast, for turn ratios exceeding 60, a clear trend emerges where the likelihood of overlapping increases with the number of

turns. The relationship between the sensing performance and nano-magnetic powder concentrations of the MC Layer is presented in Fig. S14. A semi-linear relationship was observed, with higher magnetic nanoparticle concentration generating a stronger magnetic field and, consequently, higher current output. The influence of varying PDMS ratios in the sensing membrane on the performance of the sensor is delineated in Fig. S15. An increase in the PDMS ratios was found to extend the response times and decrease the SNR while having a negligible effect on the sensitivity curve. The augmentation in PDMS ratios leads to a softer membrane, which is prone to deformation at a slower rate. Consequently, devices with higher PDMS ratios exhibit heightened sensitivity to noise-generating deformations, albeit at a reduced response time. The influence of thickness on sensing performance was tested in Fig. S16, with thicker membranes resulting in quicker response times and a fluctuating SNR. Lastly, the impact of the MC layer's thickness was tested in Fig. S17. A thicker MC layer had no influence on response time but reduced SNR. We've consolidated the results of each optimization factor in Fig. S18, providing a clear overview of the primary variables influencing each performance metric. After considering the sensing performance, weight, and flexibility of the device, the current parameters were determined. The device's durability with these parameters was evaluated in Fig. S19, where the device underwent continuous working for 24,000 cycles with a shaker under a frequency of 5 Hz, with no observable degradation in the current generation.

The acoustic performance of the actuation system of the device is examined firstly with a focus on its sound pressure level (SPL) at different distances. The results, presented in Fig. 2d, show that larger output magnification led to a higher SPL at all tested positions. Even at a distance of 1 meter, the typical distance during normal conversations, the device provided an SPL of over 40 dB, which is above the lower limit of normal speaking SPL (40–60 dB)[64]. We also tested the device's SPL at different angles and compared its performance with those of previous works on acoustic devices (Fig. S20, Supplementary Table S3). The device's performance across various frequencies was tested and presented in Fig. 2e, which indicates that it could provide sound with SPL louder than normal speaking loudness across the entire human hearing range[64]. The resonance point in the figure indicates the frequency at which the device has relatively the largest loudness output under the same signal strength as other adjacent frequencies. Further investigation into the SPL regarding frequency under different strains revealed that the first few resonance points tended to have the largest acoustic output across the frequency range (Fig. S21). Since the device under one strain has multiple resonance points that change non-linearly with deformation, investigating the change of every resonance point is complicated. Therefore, we only investigated the first resonance point (FRP) in Fig. 2f because of its complexity and our interest in the highest output. According to Fig. 2e and Fig. S22, the voice output at each strain was above the normal talking threshold across the whole human hearing range. Figure 2f revealed a right shift of FRP of the device as the deformation gets larger, enabling the device to adjust its best output performance under different usage scenarios. Our device can adjust its best output performance by simply changing the deformation degree, thus creating a unique output setting for each individual and realizing user adaptability. More details about the right shift of FRP are shown in Fig. S23.

We also tested the influence of introducing the kirigami design into the device, as presented in Fig. 2g. The results show that the parameter of the kirigami design had a negligible impact on the sensing and acoustic performance, further supporting the decision to use this design due to its impact on flexibility (Fig. S5). Additional factors influencing the acoustic performance of the actuation system were evaluated, and the final parameters were determined based on both performance and the device's mass/flexibility. Fig. S24 explores the impact of coil turn ratios on the SPL produced by the device. It was observed that an increase in coil turns led to a decrease in SPL, likely due to the weight of the additional coil impeding membrane vibration and subsequently reducing SPL. The relationship between SPL and the PDMS ratio of the actuator membrane was examined in Fig. S25. As the ratio increased, the membrane softened, leading to a decrease in the generated SPL. The dampening effect of a softer membrane hindered vibration and sound generation, resulting in a semi-linear decrease. Fig. S26 presents the relationship between SPL and magnetic powder concentrations. The device's SPL increased with the addition of higher amounts of magnetic powder in the MC layer, plateauing after a ratio of 4:1. The effect of varying MC layer thickness on SPL is shown in Fig. S27. A sharp increase in the device's SPL was observed as the MC layer's thickness increased from 0.5 mm to 1 mm. However, the increase slowed and eventually plateaued as the MC layer became thicker. Finally, the SPL under different actuator membrane thicknesses was tested in Fig. S28. The device's SPL increased as the PDMS membrane (vibrating membrane) thickness increased from 100 to 200 μm but decreased when the membrane became thicker. The weight of thicker membranes may dampen the vibration and reduce the loudness produced by the device. Regarding the acoustic output quality of the device, Fig. 2h displays the waveform of the commercial loudspeaker and our device at the maximum (164%) strain at the frequency of 1100 Hz. The device reproduced the voice signal accurately, even under maximum deformation, with only slight distortion. The distortion was further explained in the spectrogram of Fig. 2i, which shows that a noise of around 1400 Hz was generated in the output of our device but not strong enough to significantly distort the signal. Output of other strains was tested in Fig. S29, a similar distortion of less extent can be observed with less strain. In the final phase of our study, we evaluated the water resistance of our device. The waveform of the device outputting an identical voice signal segment under water and in air is depicted in Fig. S30. The waveforms are notably similar, with no significant signal distortion observed. A slight loss of the high-frequency component, without major signal attenuation, is evident in the frequency domain (Fig. S31). The device demonstrated consistent performance even after being submerged in water for an accelerated aging test with a duration of 7 days (Fig. S32). The sound pressure level (SPL) in relation to distance underwater is presented in Fig. S33. A correlation was observed between the depth of the device underwater and the sound output, with deeper submersion resulting in lower output. However, the device could produce an output exceeding 60 dB when placed 2 cm underwater at a distance of 20 cm. The SPL of the device in relation to frequency underwater is illustrated in Fig. S34. Despite the attenuation of high-frequency components underwater, the device consistently delivered an SPL above the normal speaking range (60 dB) across the entire human hearing range. These results suggest that our device, as a wearable, can effectively withstand conditions of perspiration, damp environments, and rain exposure.

## Laryngeal muscle movement signal acquisition

After obtaining the preliminary standard test results, we focused on collecting laryngeal muscle movement signals using our wearable sensing component. The experiment is schematically illustrated in Fig. 3a. The analog signal generated by the vibration of the extrinsic laryngeal muscles (Sternothyroid muscle, as shown in Fig. 3a) was collected by the sensor and then passed through an amplifier and a low-pass filter exhibited in Fig. 3b. The digital signal of the laryngeal muscle movements was output and collected for further analysis. The sensitivity and repeatability of the device were tested in Fig. 3c with two successive different throat movements. The device was able to generate distinguishable and unique signals for each different throat movement, indicating its feasibility to detect and analyze different laryngeal movement properties. Furthermore, the device responded consistently to one throat movement, as demonstrated by the

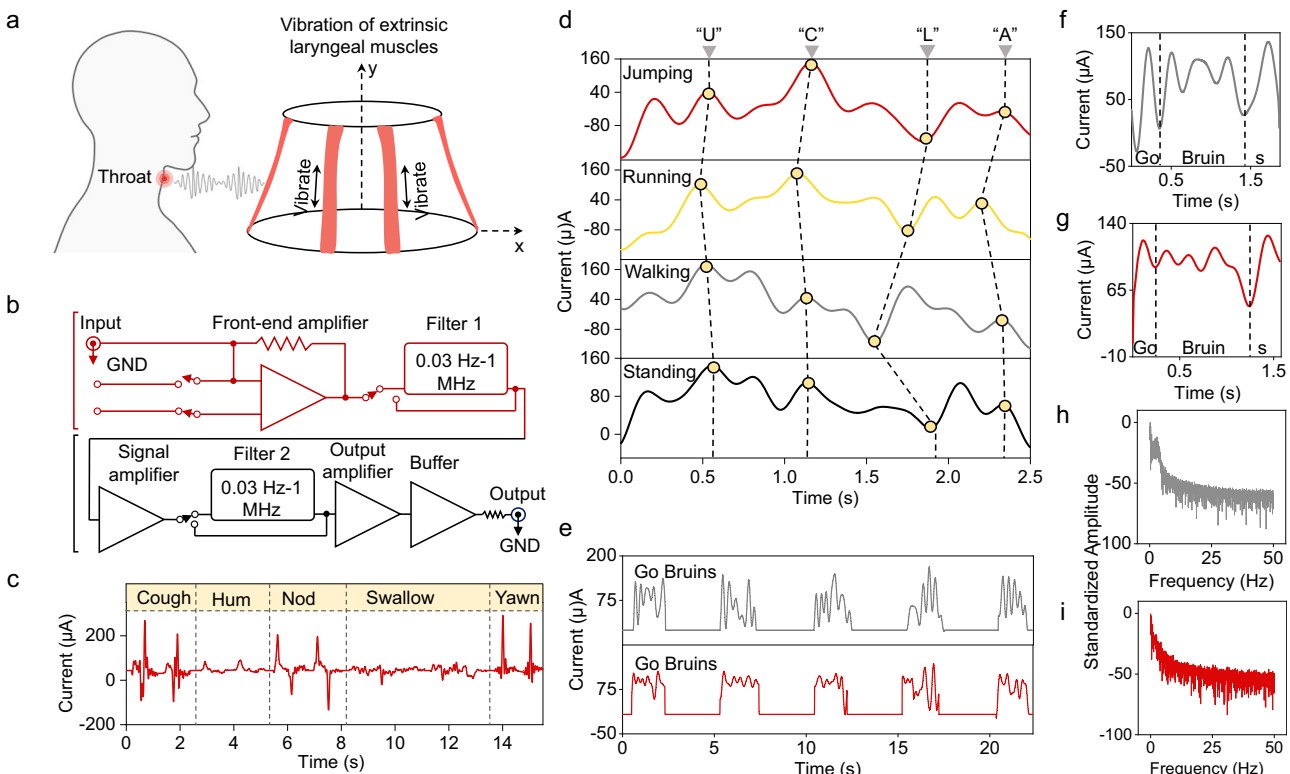

**Fig. 3 | Converting laryngeal muscle movement into analyzable electrical signals. a** Schematic illustration of the extrinsic muscle and vibration. Created with BioRender.com. **b** Circuit diagram of the system for collecting extrinsic muscle movement signal. **c** Sensor output for different throat movements—Coughing, Humming, Nodding, Swallowing, and Yawning. **d** Device signal output for participant pronouncing "UCLA" under different body movements. **e** Sensor output for participant pronouncing "Go Bruins!" with vocal fold vibration (upper, gray) and voiceless (lower, red). Enlarged waveform of participant pronouncing "Go Bruins!" with vocal fold vibration (**f**) and voiceless (**g**). Amplitude-frequency spectrum of the signal with vocal fold vibration (**h**) and voiceless (**i**).

participant's continuous two throat movements. In addition, larger throat muscle movements, such as coughing or yawning, generated larger peaks, while longer movements, such as swallowing, generated longer signals. We also conducted experiments to test the device's functionality under different conditions. In Fig. 3d, we asked the participant to voicelessly pronounce the same word ("UCLA") under different conditions, including standing still, walking, running, and jumping. The device was able to discern the unique and repeatable feature syllable wave shape of each word, with only slight differences made by the participants with different pronouncing paces each time. Thus, the wearable device was able to function without being influenced by the user's body movements, even during strenuous exercise. Finally, to test the signal quality and accuracy acquired by purely the laryngeal muscle movement, we performed examinations to compare normal speaking and voiceless speaking, as shown in Fig. 3e. The five successive signals of participant saying "Go Bruins" with and without vocal fold vibration were compared in Fig. 3f and g, respectively. Both tests generated consistent signals, and the syllables of each word were represented with distinguishable waveforms. Comparing the test results of normal speaking and speaking voicelessly, we observed only a slight loss of maximum amplitude in the signal of speaking voicelessly. This could be explained by the fact that the vibration of vocal folds requires more and stronger muscle movements, thus generating stronger signals. Furthermore, a clear loss of high-frequency components in voiceless signals compared to the signals with vocal fold vibration was observed in Fig. 3h, i after the Fourier transform of both signals across frequencies. This finding was consistent with our hypothesis that the high-frequency part of the vibration generated by intrinsic muscles and vocal folds is absent in voiceless signals, leaving a smoother yet distinguishable waveform.

Hence, the device was proven to capture recognizable and unique signals with laryngeal muscle movements for further analysis.

## Assisted speaking without vocal folds

With generated data of laryngeal muscle movement, a machine-learning algorithm was employed to classify the semantic meaning of the signal and select a corresponding voice signal for outputting through the actuation component of the system. A schematic flow chart of the machine-learning algorithm is presented in Fig. 4a. The algorithm consists of two steps: training and classifying a set of *n* sentences for which assisted speaking is required. Firstly, the filtered training data was fed to the algorithm for model training. The electrical signal of each of the *n* sentences was compacted into an *N*th-order matrix for feature extraction with principal component analysis (PCA) (Fig. 4b). *N* is determined by the sampling window, which is the length of the longest sentence's signal. PCA is applied to remove redundancy and prepare the signal for classification. Multi-class support vector classification (SVC) was chosen as the classification algorithm with the decision function shape of "one vs. rest". For each sentence to be classified, the rest of the *n-1* sentences were considered as a whole to generate a binary classification boundary to discriminate the target sentence. A brief illustration of the support vector machine (SVM) process is depicted in Fig. 4c. The margin of the linear boundary between two target data groups undergoes a series of optimizing processes and was set to the largest with support vectors. Details of PCA and multi-class SVC are discussed in Methods. After the classifier was trained with pre-fed training data, it was used for classifying newly collected laryngeal muscle movement signals. The real-time data were fed to the classifier, and the class (which sentence) of the signal was output for voice signal selection. Subsequently, the corresponding

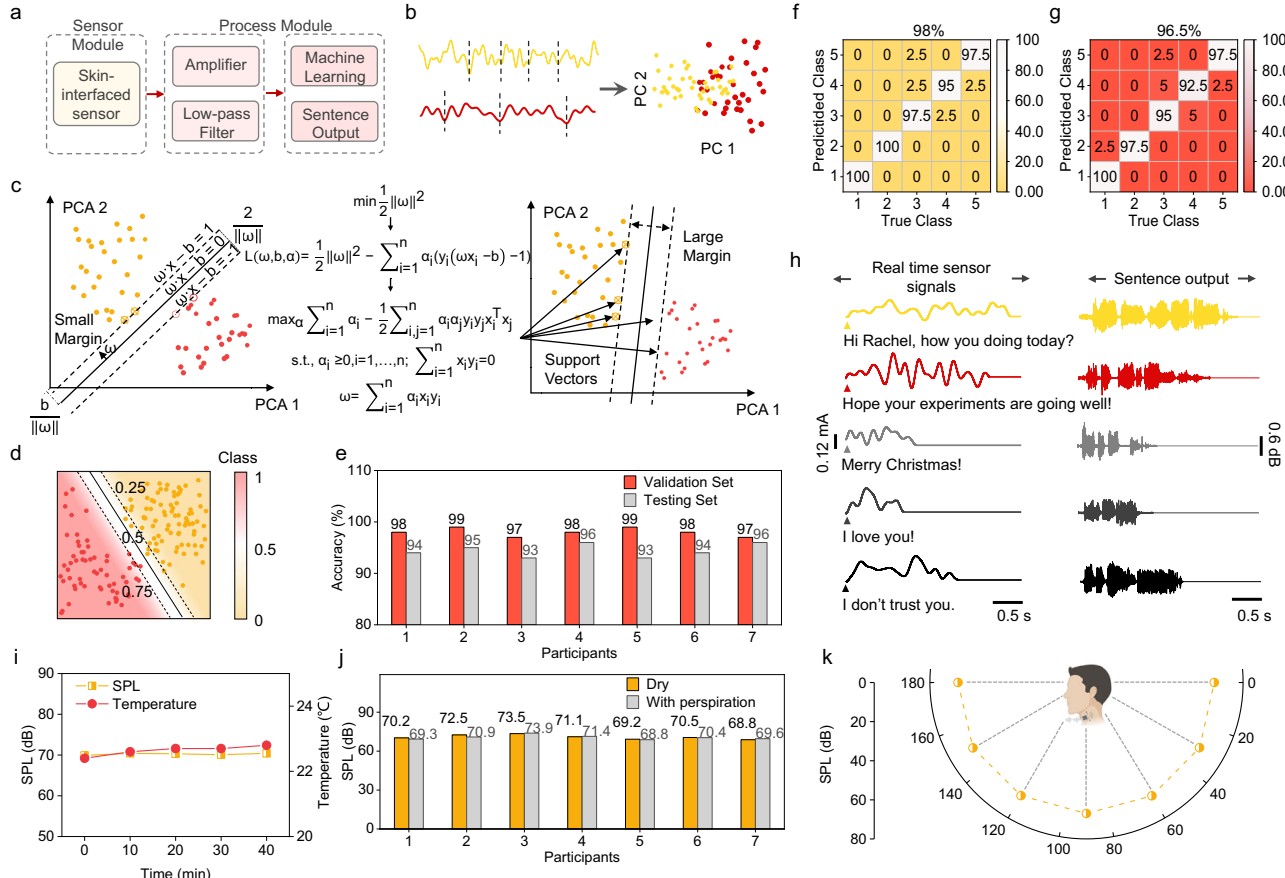

**Fig. 4 | Machine-learning-assisted wearable speaking without vocal folds.**
**a** Flow chart of the machine-learning-assisted wearable sensing-actuation system.
**b** Illustration depicting the process of data segmentation and principal components analysis (PCA) applied to the muscle movement signal captured by the sensor. Yellow indicates one sentence, and red indicates another one. **c** Optimizing process of data classification after PCA with support vector machine (SVM) algorithm. **d** Contour plot of the classification results with SVM, class "1," indicating 100% possibility of the target sentence, dotted lines are the possibility boundaries between the target sentence and the others. **e** Bar chart exhibiting 7 participants' accuracy of both validation set and testing set. **f** Confusion matrix of the 8th participant's validation set with an overall accuracy of 98%. **g** Confusion matrix of the 8th participant's testing set with an overall accuracy of 96.5%. **h** Demonstration of

the machine-learning-assisted wearable sensing-actuation system in assisted speaking. The left panel shows the muscle movement signal captured by the sensor as the participant pronounces the sentence voicelessly, while the right panel shows the corresponding output waveform produced by the system's actuation component. **i** The SPL and temperature trends over time while the device is worn by participants; no notable temperature increase or SPL decrease was seen for up to 40 min. **j** The device's SPL outputs participant-specific sound signals, both with and without sweat presence. Each participant was asked to repeat testing of $N = 3$ times for both scenarios. Data are presented as mean values ± SD. The *p*-value between dry and sweaty state is calculated to be 0.818, indicating no significant difference in the device's performance under the two cases. **k** The device's SPL across various conversation angles while done by the participant. Created with BioRender.com.

pre-recorded voice signal was played by the actuation component, realizing assisted speaking.

A brief demonstration was made with five sentences that we had selected for training the algorithm (S1: "Hi Rachel, how you are doing today?", S2: "Hope your experiments are going well!", S3: "Merry Christmas!", S4: "I love you!", S5: "I don't trust you."). Each participant repeated each sentence 100 times for data collection. The resulting contour plot in Fig. 4d shows an example of the classification result, with the red dots indicating the target sentence and the yellow dots indicating the others. A probability contour was drawn to classify whether a newly input sentence point belonged to the target sentence or not. With the trained classifier, the laryngeal movement signal was recognized for the corresponding sentence that the participant wished to express. To test the robustness and user-adaptability of the algorithm, the device was tested with eight participants, each repeating the sentence 120 times in total, with 100 repeats selected for the training set and 20 separated as the testing set. Of the 100 repeats, 20 were selected as the validation set. Figure 4e shows the validation and testing results of seven out of the eight participants, while Fig. 4f, g presents a detailed illustration of the confusion matrix of the 8th

participant for the validation and testing sets, respectively. Even slightly lower than the validation set, each participant's testing set achieved more than 93% accuracy. Figure S35 shows the detailed confusion matrix of both the validation and testing set and the accuracy of every other participant. The overall prediction accuracy of the model was 94.68%, and it worked well with different participants. Each participant's voice signal was played by the actuation component, realizing the demonstration in Fig. 4h. The left panel shows the muscle movement signal transferred into the correct voice signal, with the waveform shown in the right panel. Further, we extended our analysis to validate the practical usability of the device for vocal output after the selection of the accurate voice signal by the algorithm. As demonstrated in Fig. 4i, an evaluation of the SPL and temperature of the device during use by the participant revealed no significant drop in SPL or rise in temperature, even after an extended working period of 40 min. This suggests the device's durability in voice output and safe usage. In Fig. 4j, we display the SPL of the device as it produces voice signals for seven participants, both with and without sweat. We noted consistent performance by the device across different participants, with no evident signal attenuation despite the presence of

perspiration. Finally, Fig. 4k illustrates the device's SPL during voice output at various normal conversation angles while worn by the participant. The device demonstrated reliable sound performance across all angles, thereby enabling assisted speaking in multiple real-life scenarios. In conclusion, the device can convert laryngeal muscle movement into voice signals, providing patients with voice disorders with a feasible method to communicate during the recovery process.

## Discussion

In this work, we have developed a wearable sensing-actuation system for assisted speaking without the need for vocal folds based on magnetoelastic effects in a soft matter system. The device could translate the laryngeal muscle movement into voice signals, enabling speech without using the vocal fold. We have tested and confirmed several attractive features of the device, including a light weight of 7.2 g, high stretchability of 164%, skin-alike modulus of $7.83 \times 10^5$ Pa, a high SNR of 17.5, quick response time of 40 µs, excellent sound producing quality, and water resistance. In addition, the device has been proven to detect unique, distinguishable signals of each syllabus from the laryngeal muscle movement without losing any essential waveform characteristics for downstream analysis. With the assistance of a machine learning algorithm, the device can classify the semantic content of the movement signal and select the corresponding voice signal for outputting through the actuation component. Our device offers a compelling solution for patients with voice disorders to communicate.

## Methods

### Human subject study

In total of 8 participants were recruited in the experiment testing device performance through a questionnaire among UCLA students. Among these, 4 participants are female and 4 are male, and the average age is 21 years old. The gender information is obtained based on the self-reporting method of the participant. Gender and other biographical information are not relevant to the human study conducted in our experiment. Each participant is compensated with a gift card of $25. All participating subjects of this research are informed, and written consent of all participants was obtained before the study. The speaking without vocal folds using a machine-learning-assisted wearable sensing-actuation system was conducted in compliance with all the ethical regulations under a protocol (ID: 20-001882) that was approved by the Institutional Review Board (IRB) at the University of California, Los Angeles.

### Fabrication of the MC Layer and the kirigami structure

The neodymium–iron–boron (NdFeB, Magnequench) magnetic powder with the following properties is used in the study: Particle Size (D50), 5 µm; Residual Induction (Br): 898–908 mT, 8.98–9.08 kG; Energy Product (BH) max: 120–128 kJ/m³, 15.0–16.0 MGOe; Intrinsic Coercivity (Hci): 700–740 kA/m, 8.8–9.3 kOe; Magnetizing Field to >95% Saturation (Min.): Hs ≥ 1600 kA/m, ≥20.0 kOe; Coercive Force (Hc): 515 kA/m, 6.5 kOe. The magnetic powder is evenly mixed with polydimethylsiloxane substrate (PDMS, Sylgard 184). The PDMS is fabricated with its elastomer base and its curing agent mixed at a ratio of 15:1. Subsequently, the weight ratio of the magnetic powder and mixed PDMS is measured to be 4:1. Next, the as-prepared magnetic paste is poured into a 3D-printed mold (polylactic acid, PLA) of 30 * 30 * 1 mm (length, width, height) and transferred to an oven set at 70 °C for over 4 h. The cured MC layer was then removed from the mold and magnetized by an impulse magnetizer (IM-10-30, ASC Scientific) with an induced angle of 45° to the magnetization direction at an impulse voltage of 350 V. The magnetized MC membrane is then positioned in a laser cutter (ULTRA R5000, Universal Laser System). The desired kirigami pattern is designed using AutoCAD software and subsequently uploaded to the laser cutter. To ensure precision and depth, the laser cutter is programmed to repeatedly trace the same pattern without repositioning the MC membrane. This iterative process ensures that the cuts progressively deepen until they fully penetrate the membrane, culminating in the desired kirigami structure.

### Fabrication of serpentine-shaped-coil, sensing, and actuation membrane

A serpentine-shaped 3D printed mold is used to twine a copper coil with a diameter of 67 µm and a spacing of 22.3 ± 2.14 µm. The coil used in our final device design is 20 turns with a thickness of 147.3 µm. A sensing and actuation membrane is fabricated by scraping polydimethylsiloxane (PDMS) (10:1) onto a glass slide. The completed copper coil is then placed onto the glass slide before the membrane is cured at a temperature of 70 °C for over 4 h. The membrane is carefully removed from the glass slide with a razor blade. PDMS is then applied to the edges of the MC layer and the two membranes. The top and bottom membranes are attached to the MC layer, and the entire device is cured in an oven for another 4 h until complete.

### Electrical performance measurement

The current signal of the device is measured by a current Stanford low-noise current preamplifier (model SR570) with the following parameters, including (1) Gain Mode: We selected the "LOW NOISE" mode to ensure the most accurate and noise-free measurements. (2) Sensitivity: This was adjusted to "2 × 100 µA/V", which allowed us to capture even minute variations in the current. (3) Filter Frequency: We employed a "Lowpass 6 dB" filter set at "100 Hz". This setting was chosen to filter out any high-frequency noise that could interfere with our measurements. (4) Input Offset: This was set to "NEG" with a value of "1 × 10 µA" to account for any inherent offset in the preamplifier.

### Sweat simulation test

To evaluate the device's resilience and performance under sweaty conditions, we employed an artificial sweat surrogate (Biochemazone Inc., Artificial Sweat BZ320). The consistent composition of artificial sweat ensured uniformity across all tests. The procedure for sweat simulation and device testing includes (1) Skin Preparation: Each participant's throat area was meticulously cleaned using an alcohol pad to eliminate any natural oils or residues. After this, the area was dried with a tissue pad to ensure the complete removal of residual alcohol. (2) Initial Sweat Application: A calibrated spray bottle was utilized to evenly apply 0.5 ml of artificial sweat solution onto the cleaned skin area, simulating a layer of sweat. (3) Device Attachment: Post the artificial sweat application, the device was carefully affixed to the treated skin surface, ensuring optimal contact. (4) Secondary Sweat Application: To further mimic sweat exposure, an additional 0.5 ml of artificial sweat solution was sprayed directly onto the device's surface. (5) Settling Period: Participants were then instructed to remain stationary for a duration of 5 min. This interval was crucial to assess any potential infiltration of the artificial sweat solution into the device. (6) Data Collection: Following the settling period, the device's performance metrics were recorded under simulated sweat conditions.

### Machine-learning algorithm

Principle component analysis is used in this study to reduce the redundancy of the data and prepare data for further classification. For each throat movement signal $Xi$, it was inputted as a $N$th order matrix, where $N$ represents the longest sentence's time multiplied by the sampling point selected. In this case, $N$ equals 4 s multiplied by the sampling rate of 100, equaling 4000. The detailed theory of PCA can be found in ref. 65. Multi-class support vector machines are used in this study for classifying throat movements. The detailed theory of SVM can be found in ref. 66. SVM is a binary classification model, with its basic model being a linear classifier with the largest interval defined in the feature space. In this study, a "one vs. rest" strategy is adopted for multi-class classification. With our data set of $N$ different features

(sentences in this case), for each target feature X (X here represents the target sentence), the rest of $N-1$ features are regarded as a whole group Y. Subsequently, SVM is applied to create a linear binary between X vs Y, thus distinguishing X from the rest of the features. The same procedure is conducted for every other feature in the dataset, and a classifying boundary is set as "one vs. rest". When the data from the testing set is inputted, these boundaries are used to determine which feature this new signal belongs to, thus realizing multi-class classification.

## Statistics and reproducibility

No statistical method was used to predetermine the sample size. No data were excluded from the analyses. The experiments were not randomized. The investigators were not blinded to allocation during experiments and outcome assessment.

## Reporting summary

Further information on research design is available in the Nature Portfolio Reporting Summary linked to this article.

## Data availability

All data supporting the findings of this study are available within the article and its supplementary files. Any additional requests for information can be directed to, and will be fulfilled by, the corresponding authors. Source data are provided with this paper and can be found at DOI: 10.6084/m9.figshare.24784107. Source data are provided in this paper.

## Code availability

Codes are available from the corresponding authors upon request.

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

## Acknowledgements

The authors acknowledge the Henry Samueli School of Engineering & Applied Science and the Department of Bioengineering at the University of California, Los Angeles, for the startup support. J.C. also acknowledges the Vernroy Makoto Watanabe Excellence in Research Award at the UCLA Samueli School of Engineering, the Office of Naval Research Young Investigator Award (Award ID: N00014-24-1-2065), NIH R01 (Award ID: R01 CA287326), the American Heart Association Innovative Project Award (Award ID: 23IPA1054908), the American Heart Association Transformational Project Award (Award ID: 23TPA1141360), the American Heart Association's Second Century Early Faculty Independence Award (Award ID: 23SCEFIA1157587), the Brain & Behavior Research Foundation Young Investigator Grant (Grant No. 30944), and the NIH National Center for Advancing Translational Science UCLA CTSI (Grant No. KL2TR001882). We also acknowledge Dr. Jennifer Long, Dr. Maie St. John, Qingyan Zhou, and Jianhui Gu for providing professional insights into Otolaryngology and laryngeal anatomy.

## Author contributions

J.C. conceived the idea and supervised the whole project. Z.C. and X.W. designed the work and fabricated the device; J.X., C.D., and T.Z. assisted in the device's performance testing; Z.C., X.W., and J.C. wrote the paper and created the figure; all authors participated in the analysis of experimental data and discussion of the results.

## Competing interests

A patent has been filed related to this work from the University of California, Los Angeles with US provisional patent application No. 63/176,651.
