## [Peer Review File · Nature Communications]

REVIEWER COMMENTS

Reviewer #1 (Remarks to the Author):

The manuscript is well organized and supported by detailed results and data. The topic, machine learning used for assisted speaking through wearable sensing-actuation systems, is really interesting and the results described in the work represent a remarkable achievement in the context of the strategies to implement machine learning into advanced bioelectronics and materials engineering. This reviewer suggests the publication after some minor revisions listed below.

☒ The Introduction section should be more streamlined, especially the initial part (lines 26-45), and be more incisive in explaining the novelty and impact of the work. Is the main advantage of the proposed device the stretchability, the self-powering, the machine learning algorithm, the materials, etc.?

☒ Line 47: please explicit the acronym PVDF.

☒ Lines 46-48: the listed materials have different properties: PVDF is ferroelectric and electromechanically active, Au nanowires and graphene are electrode-like materials. The authors should briefly specify the properties of the flexible loudspeakers and the type of wearable sensors (resistive, piezoelectric, piezoresistive, etc.)

☒ Line 57: the authors should briefly introduce why they chose magnetoelasticity as the driving mechanism for the wearable platform, and possibly its advantages compared to other mechanisms (piezoelectricity, triboelectricity, piezoresistivity, etc.)

☒ Lines 75-76: the citation to Fig.1a is not clear.

☒ Line 81: how much is the thickness of the MI layer (copper coil)? And how can the author be sure that the serpentine shape of the coil “ensures the flexibility of the device while maintaining its performance”? Have they compared a non-serpentine coil with the one proposed? Or have they made preliminary EM field simulations? They should motivate a little more.

☒ Lines 84-86: the kirigami structure is used to enhance the sensitivity and stretchability, but is it applied only to the MC layer: are the other layers are also patterned in the same way or the PDMS just embodies all the inner layers?

☒ Fig S6: are the simulations related to a single layer patterned with the kirigami structure or do they consider also the contribution of the encapsulation?

☒ Lines 97-99: how are the z-axis deformations generated? Are they due to buckling effects of the narrow features connecting the kirigami units? Is the deformation in the z-direction linearly dependent on the in-plane expansion/contraction?

☒ Lines 102-109: this part is more a description of the state of the art and not results.

☒ Fig.1: the order of the panels should be revised (e-f-g)

☒ Lines 155-156: how is the number of coil turns related to the total thickness of the coil? This detail should be explicated.

☒ Lines 217-229: the waterproofness of the device was tested but only soaking tests in water until 24hours (at which temperature?). Considering the water permeability of PDMS and the other components of the device at room temperature, 24 hours could not be enough to demonstrate the device reliability. An accelerated aging test at higher temperature could be enlightening.

☒ The results in the Supplementary figures would be more digestible if the authors included some 3D summary plots with response time, SNR and sensitivity, as functions of stretching conditions, coils design, MC layer thickness, nanomagnetic powder concentration, PDMS concentration, etc.

☒ Fig.S18: the durability of the device is shown for 1700 cycles at 5Hz, i.e. 340s, which is not big enough to represent a long-term applicability. At least 60 min of operation should be enough to demonstrate the signal stability in the long term. Additionally, the durability test was performed for the unstretched device? Does the durability change if the device is already stretched at different stretching ratios? The authors can present a second durability test with a sample stretching ratio.

☒ Fig 4 is a little chaotic and hard to follow: the order of the panels should be revised.

☒ In all the paper, the self-powered feature of the proposed device does not emerge. The authors should motivate more this aspect, explaining in which sense it is self-powered: is it because of the intrinsic magnetoelastic effect of the materials?

☒ A crucial aspect to consider is whether the detected signal does not comprise also currents generated through triboelectric effects for instance due to micro-sliding motions of the MC layer within the PDMS encapsulation. The wearable application of the device can be responsible of remarkable triboelectric currents: please consider this reference (<https://doi.org/10.1002/adfm.202101047>)

☒ The conclusion section should be enriched with the most important quantitative results of the work, in terms of sensitivity, SNR, response time.

☒ Lines 318-325: how the MC layer is processed with the laser cutter? The authors should show some optical micrographs of the kirigami cuttings in order to give an idea of the sharpness of the cut edges.

Reviewer #2 (Remarks to the Author):

This manuscript presents an interesting use of magnetoelastic materials for sensing vibrations, and the application to generate voice for speech impaired people. There are scientific merits that may benefit certain sensor and medical communities. However, there are flaws in data collection, presentation and experimental procedures that render this manuscript unsuitable to be published at its current stage. The strengths of this work as well as the issues are summarized below.

Significance

Although this paper was based on previous published work, there is a clear novelty in its kirigami design of the magnetoelastic coupling layer that is interesting and also effective in increasing performance.

This technique, which uses magnetoelastic materials to detect vibrations, contains some technical innovations. However, practically, this approach may not be as impactful since piezoelectric systems can theoretically have similar performance but with a simpler design.

The authors claimed that the sensor is self-powered. It is true that although the signal is wirelessly generated by induction current, it still needs a powered device to measure the current. Thus, the wearable system will still need to be powered and the self-powered claim may be inaccurate.

Methods

Most data do not include repetitions or multiple samples. In typical sensor/device validation experiments, data are typically collected multiple times with multiple samples to ensure the measurements are consistent across different samples and are repeatable. While it makes sense that illustrative plots such as Fig 3e presents only one measurement from a single sample, performance-based plots such as Fig 2b and 2f should have data points that are the average of multiple samples (and multiple measurements per sample) with error bars indicating the standard deviation among the measurements.

The methodology used for validating the sensor is mostly sound. However, many details are missing thus making reproducing the sensor by other labs impossible. Some missing or inconsistent details such as the following should be included.

- “The entire system is small, thin (~ 1.35 ml)...” It is unclear if 1.35 ml is describing the volume of the sensor. If so, mm³ may be a better unit to use.
- In the description that the authors used magnetization impulse field to magnetize the substrate, please provide the direction and magnitude of the field.
- What is the spacing between the loops of the copper coil?
- Under Electrical Performance Measurement, the current signal of the device was measured with a preamp. Please provide the gain and filter setting used to perform the measurement.
- Please provide more information (physical and magnetic properties) of the neodymium-iron-boron (NdFeB), such as the size of the particles, the magnetization and coercivity, etc.
- Regarding the fabrication of the magnetic coupling layer, the authors stated that NdFeB magnetic powder vs PDMS mixing ratio of 15:1, but the weight ratio is 4:1? Please elaborate why there are two different mixing ratios.
- Regarding the investigation of sensor performance with or without sweat, it is unclear what is the procedure to produce sweat in the test subjects, or how much sweat on skin was being tested.

Validation experiment for the machine learning algorithm is still under very controlled situations, thus it is unclear if the sensor/AI can actually be used practically. For example, there is a lack of description on what sentences were used in the machine learning experiment, thus the accuracy of the algorithm may only be true under very narrow conditions.

Style

This manuscript can use more proof readings. There are obvious grammatical errors.

Response to Reviewers' Comments (manuscript NCOMMS-23-37367-T)

We would like to express our genuine gratitude to the reviewers for their invaluable guidance and careful evaluation of our manuscript. We are immensely grateful for the extensive recommendations made to enhance the writing style and refine the experimental details, which have effectively expanded upon our arguments and bolstered the overall quality of the manuscript. The insightful suggestions and subsequent revisions have significantly improved the clarity of our work, unified our endeavors, and enhanced its accessibility for future readers.

Reviewer #1:

The manuscript is well organized and supported by detailed results and data. The topic, machine learning used for assisted speaking through wearable sensing-actuation systems, is really interesting and the results described in the work represent a remarkable achievement in the context of the strategies to implement machine learning into advanced bioelectronics and materials engineering. This reviewer suggests the publication after some minor revisions listed below.

Response:

We are extremely grateful to the reviewer for acknowledging our work in the field of advanced bioelectronics and materials engineering. We deeply appreciate the reviewer's encouragement and valuable remarks, including phrases such as, "*really interesting*" and "*remarkable achievement*".

1. The Introduction section should be more streamlined, especially the initial part (lines 26-45), and be more incisive in explaining the novelty and impact of the work. Is the main advantage of the proposed device the stretchability, the self-powering, the machine learning algorithm, the materials, etc.?

Response:

We are grateful to the reviewer for their constructive feedback on the Introduction section of our manuscript. We recognize the importance of clearly conveying the novelty and impact of our work from the outset.

In response to the reviewer's query about the primary advantage of our device, we have refined the Introduction part to emphasize the unique attributes and challenges addressed by our proposed solution. Specifically, we have highlighted the limitations of existing assisted speaking solutions, such as their non-perspiration-proof nature, inability to capture parallel muscle deformation, and lack of stretchability. The revised sections in the main text read:

"However, the non-stretchable nature of these materials has limited their application, as they primarily capture vertical throat movement, overlooking the parallel deformation during phonation involving various laryngeal muscle groups"

"And other issues of those materials such as lack of water (perspiration) resistance and temperature rise can lead to operational problems."

The standout advantage of our device lies in the choice of material. Its inherent stretchability allows for a comprehensive capture of three-dimensional throat muscle movements. Additionally, the magnetic-based working principle ensures resistance to perspiration. These material properties address the aforementioned challenges and form the crux of our contribution.

To further elucidate the primary advantage and impact of our work, we have added the following to the main text:

“Furthermore, based on the nature of the flexible substrate used, the system's significant stretchability (164%) for horizontal deformation detection enhances adhesive attachment to the throat, contributing to comfort and precise movement detection, tackling the crucial issue of capturing omnidirectional mechanical deformation. The system is intrinsically waterproof since magnetic field is not attenuated by water, ensuring durability and functionality even in the presence of factors like perspiration.”

We trust that these revisions provide a clearer and more incisive understanding of the novelty and significance of our research.

2. Please explicit the acronym PVDF.

Response:

We would like to express our deep appreciation to the reviewer for pointing out the acronym used in the article. We have carefully gone through all the abbreviations used in the manuscript to make sure the clear expressions. Adjustments have been made in the main text.

“polyvinylidene fluoride (PVDF)”

3. The listed materials have different properties: PVDF is ferroelectric and electromechanically active, Au nanowires and graphene are electrode-like materials. The authors should briefly specify the properties of the flexible loudspeakers and the type of wearable sensors (resistive, piezoelectric, piezoresistive, etc.)

Response:

We are grateful to the reviewer for highlighting the need for clarity regarding the properties of the materials used in our study. It is essential to provide a comprehensive understanding of the materials' characteristics to appreciate their relevance in the context of our work.

In response to the reviewer's feedback, we have elaborated on the properties of the materials in the introduction:

“PVDF emerges as a pristine thermoplastic fluoropolymer, notable for its exceptional non-reactivity. A distinguishing feature of PVDF is its piezoelectric property, adeptly converting mechanical oscillations into precise voltage signals. While this piezoelectric property offers certain advantages, the material selection for piezoelectric sensors remains limited, often constraining the design and functionality of devices tailored for specific applications. In parallel, gold nanowires and graphene have gained recognition for their superior conductivity and inherent flexibility. These characteristics make them ideal candidates for crafting resistive sensors, which can swiftly measure resistance changes in response to mechanical stresses. However, these resistive sensors, including those made from gold nanowires, typically require an external power source for sensing, adding to the complexity and potential bulkiness of the wearable system.”

4. The authors should briefly introduce why they chose magnetoelasticity as the driving mechanism for the wearable platform, and possibly its advantages compared to other mechanisms (piezoelectricity, triboelectricity, piezoresistivity, etc.)

Response:

We greatly appreciate that the reviewer brought this to our attention. The system we have developed, based on magnetoelasticity, encompasses **both sensing and actuation components**. It is designed to detect biomechanical motion signals and subsequently produce voice signals *via* an actuation component, in this case, a loudspeaker. The soft substrate inherent to magnetoelastic materials ensures that both the sensing and actuation components are flexible and stretchable, making them ideal for wearable applications. Additionally, the system's low driving voltage enhances the safety of the wearable devices. While there are alternative materials that can be used for sensing, they may not offer comparable performance in actuation as magnetoelastic materials. For example, piezoelectric materials are commonly used for mechanical-electrical energy conversion. However, their driving

voltage for actuation tasks, such as in ultrasound probes, is relatively high. This could raise safety concerns when integrated into wearable bioelectronics. Also, as we mentioned in the previous part, certain issues exist in the concurrent solutions for assisted speaking as non-perspiration-proof, overlooking the parallel deformation of muscles, and non-stretchability. **The stretchability, self-powering sensing feature, durability and waterproofness of the magnetoelasticity materials used in our work tackles those issues correspondingly.** This attributes to the stretchable substrate we used and non-blocking of magnetic field by water. We have first pointed out the issues with the existing materials in the second paragraph of the Introduction:

“However, the non-stretchable nature of these materials has limited their application, as they primarily capture vertical throat movement, overlooking the parallel deformation during phonation involving various laryngeal muscle groups”

“Also, even though piezoelectrical materials present actuation abilities, the driving voltage would induce safety concerns for wearable bioelectronics.”

“And other issues of those materials such as lack of water resistance and temperature rise can lead to operational problems.”

And subsequently discussed how each of those issues can be solved by the magnetoelasticity material we used in this work as follows:

“Furthermore, based on the nature of the flexible substrate used, the system's significant stretchability (164%) for horizontal deformation detection enhances adhesive attachment to the throat, contributing to comfort and precise movement detection, tackling the crucial issue of capturing omnidirectional mechanical deformation. The magnetoelastic property of the material enables both sensing and actuation in one soft and stretchable system. The system is intrinsically waterproof since magnetic field is not attenuated by water, ensuring durability and functionality even in the presence of factors like perspiration. Beyond the materials currently utilized, a broad spectrum of magnetic particles, polymers and hydrogels can theoretically serve as the soft substrate, offering a vast array of material choices.”

5. The citation to Fig. 1a is not clear.

Response:

We greatly appreciate the reviewer's attention to detail and his/her question regarding the citation of Fig. 1a. Corresponding adjustments have been made in the main text.

*“For speaking without vocal folds, **Fig. 1a** illustrates a thin, flexible, and adhesive wearable sensing-actuation system attached to the throat surface.”*

6. How much is the thickness of the MI layer (copper coil)? And how can the author be sure that the serpentine shape of the coil “ensures the flexibility of the device while maintaining its performance”? Have they compared a non-serpentine coil with the one proposed? Or have they made preliminary EM field simulations? They should motivate a little more.

Response:

We are very much grateful for the careful attention to detail of the reviewer regarding the question of the MI layer of the device. Thank you for your meticulous observation concerning the MI layer of our device. The coil's serpentine shape is crafted by wiring coil circles concentrically from the center outward, using a serpentine-shaped mold, as depicted in **Figure R1**. While the coil is layered sequentially on the xy-plane, its theoretical thickness should equate to the diameter of the copper coil, which is 67 μm . However, during the fabrication process, the wire tends to overlap, even with efforts to flatten it. We assessed the thickness of several coils (a total of six) used in our device and found an average thickness of 147.33 μm (with individual measurements of 144,

156, 141, 139, 159, and 145 μm). This suggests that, on average, two layers of copper wire overlap during fabrication, leading to an increased overall coil thickness.

Fig. R1 | Illustration of the MI layer. It consists of fine copper wires.

In order to further ensure the performance of the coil compared to the non-serpentine-shaped coils, we have conducted the following experiments. In **Figure R2a**, we tested a serpentine-shaped coil with 20 turns and an outermost side length of $L = 3$ cm using a shaker operating at 5 Hz. Aside from a minor fluctuation in signal amplitude at the test's onset, a consistent 5 Hz response signal with an amplitude of approximately $13 \mu\text{A}$ was observed. Using a similar experimental setup, we tested a non-serpentine-shaped coil with 20 turns. The results, presented in **Figure R2b**, show a comparable response waveform with an amplitude of roughly $12 \mu\text{A}$. We calculated the Signal-to-Noise ratio for both scenarios, obtaining values of 23.4 dB and 24.6 dB, respectively.

The similarity in performance can be attributed to the current generation principle of our device, which is described by the equation:

$$\varepsilon = -N \frac{\Delta\varphi}{\Delta t}, \varphi = BS$$

Where ε represents the induced electromotive force, N is the number of turns in the coil, φ is the magnetic flux, and t is time, B is the magnetic field intensity, and S is the surface area of the coil.

According to the law of electromagnetic induction, the generated voltage in the circuit is proportional to the coil turns and the rate of change in magnetic flux within the area enclosed by the coil turns. In the cases we examined, the change in magnetic field, ΔB , is produced by the same MC layer that underwent identical deformations, strictly controlled by manipulating the shaker setting (including frequency, amplitude, and strength). In this case, the only variable is S , the coil's area. Given our experimental conditions and the consistent outermost side length, both coils possess nearly identical areas, with minor variations resulting from the serpentine shaping. Consequently, the outputs of both coils are closely matched.

Fig. R2 | Testing of signal generated by two different shaped coils. **a**, Signal generated by the serpentine-shaped coil and the illustration of the serpentine-shaped-coil. $L = 3$ cm in the illustration. **b**, Signal generated by the non-serpentine-shaped coil and its illustration. $L = 3$ cm in the illustration.

We have also added a supplementary note providing the detailed discussion regarding the performance of the serpentine-shaped coil. Corresponding changes have also been made in the main text:

“The serpentine shape of the coil ensures the flexibility of the device while maintaining its performance, as shown in the detailed discussion in Supplementary Note 1.”

7. The kirigami structure is used to enhance the sensitivity and stretchability, but is it applied only to the MC layer: are the other layers are also patterned in the same way or the PDMS just embodies all the inner layers?

Response:

We would like to express our deep gratitude to the reviewer for raising the question concerning the fabrication of both MC and MI layers. Our design incorporates two distinct layers: the MC layer and the MI layer. Once the serpentine-shaped coil is crafted, it is placed into a mold containing uncured PDMS. Upon curing, the coil becomes embedded within the PDMS membrane, forming the MI layer, which serves both sensing and actuating functions. The kirigami structure is exclusively implemented in the MC layer for two primary reasons:

1. **The reduced thickness of the MI layer inherently provides it with stretchability**, negating the need for a kirigami design. Through optimization, as illustrated in Figures S16 and S17, we established the thicknesses of the MI and MC layers at 200 μm and 1000 μm , respectively. Given that both layers utilize PDMS as their foundational substrate, our tests revealed that the thicker MC layer is more resistant to deformation and possesses a lower maximum strain compared to the MI layer. While the kirigami design enhances the MC layer's maximum strain to 164%, the MI layer still exhibits superior stretchability, exceeding 180%. Thus, the MC layer's stretchability becomes the limiting factor for the overall device's extensibility.

Figure R3 | Response time, SNR and sensitivity of the device with regards to different thickness of the sensing membrane and the MC layer. a, the response time and SNR regarding PDMS thickness. Thicker membrane results in quicker response times and a fluctuating SNR. **b,** the sensitivity under different PDMS ratios.

Only a slight decrease on the sensitivity has been caused by the PDMS thickness. With the results, we have determined the thickness in our design to be 200 μm . Even though higher thickness has quicker response time, the thicker membrane will affect the wearing comfort and adherence of the device attached to the skin. **c**, the response time and SNR regarding the PDMS thickness. Thicker MC layer has no influence on response time while reduces SNR. **d**, the sensitivity under different MC layer thickness. An even increase on the sensitivity can be observed. Since thicker MC layer creates stronger magnetic field, the current generated at each amplification level is larger. Similarly, the noise signal is also stronger, thus, reducing the SNR.

2. **The MC layer is solely tasked with detecting the three-dimensional mechanical signals associated with muscle movements.** Another objective of the kirigami design is to accurately capture the tri-dimensional dynamics of the laryngeal muscle. As the MC layer is responsible for translating bio-mechanical movements into changes in magnetic flux, it is crucial to ensure precise conversion at this stage.

To address the reviewer's kind concerns, we have added the Figure R3 as the new Supplementary Figure S16.

8. Are the simulations related to a single layer patterned with the kirigami structure or do they consider also the contribution of the encapsulation?

Response:

We would like to express our deep gratitude to the reviewer for raising the question concerning the simulation of the MC layer. The simulation is related to only the fabricated MC layer. As discussed in the previous question, the MC layer is the limiting factor of both the stretchability and the detecting performance since the PDMS encapsulation has higher maximum strain. So, elevating the stretchability of MC layer also elevates the overall stretchability of the device.

9. How are the z-axis deformations generated? Are they due to buckling effects of the narrow features connecting the kirigami units? Is the deformation in the z-direction linearly dependent on the in-plane expansion/contraction?

Response:

We thank the reviewer for bringing this detail to our attention. The deformation along the z-axis arises from the expansion of the muscle bundle's diameter during its contraction. As illustrated in **Figure R4a**, in a relaxed state, the muscle bundle's diameter (along the z-axis) diminishes while its length (in the x-y plane) extends. Conversely, during the contraction phase, as depicted in **Figure R4b**, the muscle bundle's length contracts, leading to an increase in its diameter. The cross-sectional views of the device, when affixed to the muscle bundle in these two states, are presented in **Figures R4c** and **R4d**. In the relaxed state, the device adheres to the muscle surface with minimal curvature, predominantly lying in the x-y plane. However, as the muscle contracts and its diameter expand, the device assumes a curved profile on the muscle surface, resulting in a z-axis deformation, denoted as D_z . This deformation, in contrast to a purely in-plane shift, imparts a "bending" effect on the device. The kirigami fabrication lowers the modulus of the device to skin-alike level, ensuring synchronized deformation with the human body, thus capturing every mechanical movement of the laryngeal muscle.

In order to further determine the relationship between in-plane expansion/ contraction and the z-axis deformation, we have modeled the muscle contraction process in **Figure R4e**. We have modeled the cross-section of the muscle bundle a semi-circle with a radius R . And the device shown in red is attached to the surface of the circle, the length of the device can be calculated with arc length formula, where θ is the central angle:

$$L = \theta \cdot R$$

The z-axis deformation generated D_z can be calculated by the radius R minus its cosine value of half θ to be:

$$D_z = R - R \cos\left(\frac{\theta}{2}\right)$$

Since the device is adhesively attached to the skin surface, the relative position between the device and skin does not change during deformation. Under minor deformation, we can set θ as constant, in this case in-plane deformation dL can be calculated as:

$$\Delta L = \theta \cdot \Delta R$$

And:

$$\Delta R = \frac{\Delta L}{\theta}, \Delta Dz = \frac{\Delta L}{\theta} \left(1 - \cos\left(\frac{\theta}{2}\right)\right)$$

So, the deformation in the z-direction linearly dependent on the in-plane expansion/contraction in a minor scale. When the deformation extend increases the θ in not a constant since the circle changes into an oval, **the relationship between z-axis deformation and the in-plane expansion/contraction involves a non-constant trigonometric function, thus not linear.** While the relationship between Dz and L, θ is not linear, it is nonetheless monotonic, as indicated by the formula provided. This ensures that each distinct laryngeal muscle movement will produce a unique deformation in the device, which in turn generates a unique and identifiable electrical signal for downstream processing. In simpler terms, each specific laryngeal movement is represented by a unique electrical waveform captured by the device's sensing component, thereby guaranteeing the device's sensing accuracy and performance.

We have also added a supplementary note providing the detailed discussion of the relationship between the omnidirectional laryngeal movements and the device' sensing performance. Corresponding changes have also been made in the main text:

“The Supplementary Note 2 further demonstrates the response of the device to the omnidirectional laryngeal movements and how the kirigami structure ensures the sensing performance.”

Fig. R4 | The relationship between z-axis deformation and the in-plane expansion/contraction. **a**, illustration of muscle bundle during relaxation. **b**, illustration of muscle bundle during contraction. **c**, cross-section view of the device attached to the muscle bundle during relaxation. **d**, cross-section view of the device attached to the muscle bundle during contraction. **e**, geometric model of the relationship between z-axis deformation and the in-plane expansion/contraction.

10. Lines 102-109: this part is more a description of the state of the art and not results.

Response:

We appreciate the reviewer for bringing up this critical question. We have tested the magnetomechanical coupling factor of the system ($7.17 \times 10^{-8} \text{ T Pa}^{-1}$, four times higher than conventional materials) adopted in our work and have made comparison with conventional rigid metal alloys. Further validating the feasibility of choosing magnetoelastic effect based on soft composite system as the platform technology of the biosensor used in our work. Corresponding adjustments have been made in the article:

“Historically, these materials received limited attention within the bioelectronics domain for several reasons: the magnetization variation of magnetic alloys within biomechanical stress ranges is constrained; the necessity for an external magnetic field introduces structural intricacies; and a significant mechanical modulus mismatch exists between magnetic alloys and human tissue, differing by six orders of magnitude. However, a breakthrough occurred in 2021 when the pronounced magnetoelastic effect was identified in a soft composite system⁵⁴. This system exhibited a peak magnetomechanical coupling factor of $7.17 \times 10^{-8} \text{ T Pa}^{-1}$, representing an enhancement up to fourfold compared to traditional rigid metal alloys, underscoring its potential in bioelectronics.”

11. The order of the panels should be revised (e-f-g)

Response:

We appreciate the reviewer's suggestions and recognize the sequences of the panels in the figure. And we have revised the sequence accordingly as follows, changes have also been made in the main text:

Fig. R5. Revised Fig. 1 | Design of the wearable sensing-actuation system. a, Illustration of the wearable sensing and phonation system attached to the throat. **b**, Explosion diagram exhibiting each layer of the device

design. **c**, Two modes of the muscle movement, expansion induces the elongation in the x and y axis, while contraction induces the elongation in the z axis. Kirigami-structured device response to muscle movement patterns in the x, y (**d**) and z direction (**e**): expansion results in x and y axis expansion and less deformation in the z axis, contraction results in less deformation in x and y direction and expansion of the z axis. **f**, Detailed illustration of the magnetic field change caused by magnetic particles. For one part, the angle change between each single unit of the kirigami structure is represented by ϕ . For the other part, the magnetic particle itself undergo torque caused by the deformation applied onto the polymer (**g**), thus, generating a change of magnetic flux and subsequently current in the coil. The photo of the device in muscle expansion state is shown in **h** (x, y axis), **i** (z axis) and in muscle contraction state is shown in **j** (x, y axis), **k** (z axis). Scale bars, 1cm.

12. How is the number of coil turns related to the total thickness of the coil? This detail should be explicated.

Response:

We thank the reviewer for his/her concerns and constructive feedback. As addressed in our earlier response, the theoretical thickness of the coil is equivalent to the diameter of the copper wire utilized in our study, which is 67 μm . Nevertheless, during the actual fabrication, the copper wires have a propensity to stack atop one another, thereby augmenting the overall coil thickness. Drawing a parallel to the experiment referenced in question 6, we constructed coils with varying turns and tabulated the results as follows:

Table R1. Thickness of the coil with different turn ratios.

Sample	20 Turns (μm)	40 Turns (μm)	60 Turns (μm)	80 Turns (μm)	100 Turns (μm)
1	144	140	147	154	163
2	156	137	155	201	201
3	141	146	161	199	270
4	139	151	153	160	204
5	159	155	198	207	201
6	145	151	143	210	211

Referencing Table R1, it's evident that as the number of coil turns escalates, there's a direct correlation with the likelihood of copper wires stacking. Consequently, a significant number of samples exhibit thicknesses approximating 2 or 3 layers of copper (134 μm and 201 μm , respectively). This stacking effect amplifies the average coil thickness as the number of turns increases. However, this augmentation isn't strictly linear. For instance, the propensity for overlapping is less pronounced for turn ratios of 20 and 40. In contrast, for turn ratios exceeding 60, a clear trend emerges where the likelihood of overlapping increases with the number of turns.

To address the reviewer's concern, we have added the Table R1 to the supplementary information as the Supplementary Table S2, and made changes in the main text as follows:

Results:

“We have further investigated the increase of thickness with the coil turn ratios in Supplementary Table S2. As the number of coil turns escalates, there's a direct correlation with the likelihood of copper wires stacking. Consequently, a significant number of samples exhibit thicknesses approximating 2 or 3 layers of copper (134 μm and 201 μm , respectively). This stacking effect amplifies the average coil thickness as the number of turns increases. However, this augmentation isn't strictly linear. For instance, the propensity for overlapping is less pronounced for turn ratios of 20 and 40. In contrast, for turn ratios exceeding 60, a clear trend emerges where the likelihood of overlapping increases with the number of turns.”

Methods:

“A serpentine-shaped 3D printed mold is used to twine a copper coil with a diameter of $67\ \mu\text{m}$ and a spacing of $22.3 \pm 2.14\ \mu\text{m}$. The coil used in our final device design is 20 turns with a thickness of $147.3\ \mu\text{m}$.”

13. The waterproofness of the device was tested but only soaking tests in water until 24 hours (at which temperature?). Considering the water permeability of PDMS and the other components of the device at room temperature, 24 hours could not be enough to demonstrate the device reliability. An accelerated aging test at higher temperature could be enlightening.

Response:

We would like to extend our sincere appreciation to the reviewer for raising the insightful question regarding the waterproofness of the device. To address this, we undertook additional experiments to rigorously assess its waterproofness. The device was immersed in PBS (Phosphate-buffered saline) and subsequently placed in an oven maintained at a consistent temperature of 65°C . We evaluated the sound pressure level (SPL), response time, and signal to noise ratio (SNR) of the device at 24-hour intervals, continuing up to 168 hours. The findings are presented in **Figure R6**. As depicted in Figure R6a, there is no discernible degradation or disintegration of the device. Furthermore, Figure R6b reveals that the response time, SPL, and SNR of the device remained largely stable, with only minor fluctuations (add data here, e.g., how much percentage...) observed. This consistency underscores the device's robust waterproof attributes, even under the conditions of an accelerated aging test. Corresponding changes have also been made in the supplementary information as supplemental Figure 32.

Fig. R6 | Accelerated aging test of the device for waterproofness. a, Photos of the device at day 1 and day 7. Scale bar, 0.75 cm. **b,** SPL, Response time, and SNR of the device after different soaking hours.

14. The results in the Supplementary figures would be more digestible if the authors included some 3D summary plots with response time, SNR and sensitivity, as functions of stretching conditions, coils design, MC layer thickness, nanomagnetic powder concentration, PDMS concentration, etc.

Response:

We greatly appreciate the valuable suggestions of the reviewer. We have created the following 3D summary plots with response time, SNR, and sensitivity, as functions of stretching conditions, coils design, MC layer thickness, nanomagnetic powder concentration, PDMS concentration. As shown in **Figure R7**, the influence of each of the optimizing factors on Response time, SNR, and sensitivity is shown in Figure R7a, R7b, R7c respectively. With the figure, the device optimizing results are much more digestible and it is obvious to observe which parameter (s) affects the corresponding performance the most. We thank the reviewer again for providing this insightful suggestion and corresponding changes have also been made in the supplementary information and the main text:

“We have summarized the outcomes of each optimization factor in Fig. S18, offering a comprehensive insight into the primary variables affecting each performance metric. The predominant variable for response time is the Coil Turn ratios. For SNR, the primary influencing factors are the PDMS Ratios and Coil Turn Ratios, with these

two factors exerting opposing effects. Meanwhile, the sensitivity is majorly influenced by the thickness of the MC layer and the Coil Turn Ratios.”

Fig. R7 | 3D Summary plots of the optimizing parameters’ influence on Response time, SNR, and Sensitivity. **a**, the Response time of the device under different parameters of coils ratio, MC layer thickness (ratio of normalized scale), nanomagnetic powder concentration (%), MI layer thickness (ratio of normalized scale), and PDMS ratio. **b**, the SNR of the device under different parameters of coils turns (turns), MC layer thickness (μm), nanomagnetic powder concentration, MI layer thickness (μm), and PDMS ratio. **c**, the Sensitivity of the device under different parameters of coils turns (turns), MC layer thickness (μm), nanomagnetic powder concentration, MI layer thickness (μm), and PDMS ratio.

15. The durability of the device is shown for 1700 cycles at 5Hz, i.e., 340s, which is not big enough to represent a long-term applicability. At least 60 min of operation should be enough to demonstrate the signal stability in the long term. Additionally, the durability test was performed for the unstretched device? Does the durability change if the device is already stretched at different stretching ratios? The authors can present a second durability test with a sample stretching ratio.

Response:

We are grateful to the reviewer for their keen observations and inquiries regarding the device's durability. In response to the suggestions, we conducted further experiments. **Figure R8a** illustrates the device being tested using a shaker at a frequency of 5 Hz for a duration of 80 minutes, totaling 24,000 cycles. The impact area of both the stretched and unstretched device was set to be identical with a 3D-printed rod to be 3 cm², the outputting pressure of the shaker was set to be identical through identical amplifier output power of 0.5 a.u. Figures R8b and R8c provide a magnified view of the waveforms from cycles 115-140 and 24,053-24,073, respectively. These figures clearly indicate that there is no significant attenuation of the signal over time, and the current amplitude remains consistent. We also tested the device under similar conditions but with a stretching degree of 135% strain, as depicted in Figure R8d. The magnified waveforms for this test, from cycles 115-140 and 24,053-24,073, are presented in Figures R8e and R8f. Once again, we observed no attenuation or change in current amplitude from the start to the end of the experiment. There was only a slight decrease in current amplitude when compared to

the unstretched device. We have updated the main text and the supporting information (Fig. S19) to reflect these findings.

Fig. R8 | Durability test of the device. **a**, the overall waveform of the device tested without being stretched from 1-24,000 cycles. **b**, enlarged view of the 115-140 cycles of the device. **c**, enlarged view of the 24050-24075 cycles of the device. **d**, the overall waveform of the device tested under 135% strain from 1-24,000 cycles. **e**, enlarged view of the 115-140 cycles of the stretched device. **f**, enlarged view of the 24050-24075 cycles of the stretched device.

16. Fig 4 is a little chaotic and hard to follow: the order of the panels should be revised.

Response:

We sincerely thank the reviewer for suggesting the order of the panels in Figure 4. We have adjusted the figure as follows to ensure the readability and the logical sequence. Corresponding changes has also been made in the main text.

Fig. R9. Revised Figure 4 | Machine-learning-assisted wearable speaking without vocal folds. **a**, Flow chart of the machine-learning-assisted wearable sensing-actuation system. **b**, Illustration depicting the process of data segmentation and Principal Components Analysis (PCA) applied to the muscle movement signal captured by the sensor. **c**, Optimizing process of data classification after PCA with Support Vector Machine (SVM) algorithm. **d**, Contour plot of the classification result with SVM, class "1" indicating 100% possibility of the target sentence, dotted lines are the possibility boundaries between the target sentence and the others. **e**, Bar chart exhibiting 7 participants' accuracy of both validation set and testing set. **f**, Confusion matrix of the 8th participant's validation set with an overall accuracy of 98%. **g**, Confusion matrix of the 8th participant's testing set with an overall accuracy of 96.5%. **h**, Demonstration of the machine-learning-assisted wearable sensing-actuation system in assisted speaking. The left panel shows the muscle movement signal captured by the sensor as the participant pronounces the sentence voicelessly, while the right panel shows the corresponding output waveform produced by the system's actuation component. **i**, The SPL and temperature trends over time while the device is worn by participants; no notable temperature increase or SPL decrease was seen for up to 40 minutes. **j**, The device's SPL as it outputs participant-specific sound signals, both with and without sweat presence. **k**, The device's SPL across various conversation angles while donned by the participant.

17. In all the paper, the self-powered feature of the proposed device does not emerge. The authors should motivate more this aspect, explaining in which sense it is self-powered: is it because of the intrinsic magnetoelastic effect of the materials?

Response:

We feel so grateful to the reviewer for their feedback regarding the "self-powered" claim. By "self-powered," we mean that the device generates an electrical signal in response to the biomechanical vibrations of the laryngeal muscles, without the need for an external power source during the sensing process. The MC layer captures this biomechanical movement and converts it into an electrical signal through the magnetoelastic effect and electromagnetic induction. This conversion represents an electromechanical energy exchange and is inherently a current-generating process. Unlike some sensing mechanisms, such as resistive sensors that necessitate an external circuit for signal detection, our sensing component operates in a self-powered manner. While additional power management circuits are essential for processing and filtering the signal, the initial sensing phase is autonomous and does not rely on an external energy supply. Corresponding changes have also been made in the main text:

“This conversion represents an electromechanical energy exchange and is inherently a current-generating process. Unlike some sensing mechanisms, such as resistive sensors that necessitate an external circuit for signal detection, our sensing component operates in a self-powered manner. While additional power management circuits are essential for processing and filtering the signal, the initial sensing phase is autonomous and does not rely on an external energy supply, thus realizing self-powered biomechanical signal collection.”

18. A crucial aspect to consider is whether the detected signal does not comprise also currents generated through triboelectric effects for instance due to micro-sliding motions of the MC layer within the PDMS encapsulation. The wearable application of the device can be responsible of remarkable triboelectric currents: please consider this reference (<https://doi.org/10.1002/adfm.202101047>).

Response:

We greatly appreciate the reviewer's astute inquiry about our study. After thoroughly examining the reference shared by the reviewer, we identified several key factors necessary for the electrical conversion of TENG: 1. A contact separation gap must exist between the two layers of materials. 2. A single electrode TENG must be grounded to facilitate charge exchange with the ground. 3. The current typically remains at the μA level, stemming from TENG's capacitive operating mechanism. However, our MEG does not require these three conditions to produce electrical signals with much higher current signal at the mA level. We delve into this topic further and present detailed experimental results below:

1. **Difference in structure design and operation mode.** A typical MEG holds an all-in-one body design, there is no relative movement among layers. Furthermore, an adhesive layer was applied to prevent any potential sliding between the MEG device and the skin. As depicted in Fig. R10a, our MEG device is securely affixed to the skin with an adhesive layer, which prevents the relative sliding/movement between the MEG device and skin. The sensing component consists of the MC layer and the MI layer, both of which are fully embedded in the PDMS matrix with an all-in-one body design. To ensure a secure attachment, we utilized adhesive to affix the device to the skin. This ensures that there is no relative movement between the skin and the MEG device.

Fig. R10 | Device structure comparison between MEGs and Single-Electrode TENGs. **a**, illustration of device structure of MEGs. There is no relative movement between skin and MEG device with a strong adhesive layer in between for operation. **b**, illustration of device structure of TENGs with single electrode mode.

2. **Difference in electrode numbers and connection.** The MEG has two electrode outlets that were connected to the two ends of the loading resistance, while TENG only has one electrode outlet, going through the loading resistance, and then connected to the ground. The electrode connection in the circuit makes our device unable to satisfy the structure of a TENG. We have thoroughly reviewed the reference you kindly provided. To address the micro-sliding motions discussed therein, we draw attention to the single electrode mode of TENG illustrated in Fig. R10b. In contrast to the other three TENG working modes (contact-separation, sliding, and free-standing) which necessitate two electrodes, our setup involves only the MI layer connected to the coil, potentially acting as a TENG electrode. For the single electrode TENG mode, one circuit end is linked to an electrode that continuously interacts with another material possessing a different electronegativity, while the other end is grounded. This facilitates electron flow to the ground, producing a current. In our design, both electrode ends connect to a current meter or load, making it implausible to replicate the single electrode TENG mode.

3. **Order of magnitude difference in electrical output.** The typical current output of a MEG is at the mA level, while the TENG is at the at the μA level. The typical voltage output of a MEG is at the 100 mV level, while the TENG is at the at the 100 V level. The typical inner impedance of a MEG is at the 10 Ω level, while the TENG is at the at the M Ω level. The high current output excludes the potential of current generation from triboelectric effect. Typically, TENG's current generation is minimal due to its elevated internal impedance. Conversely, the magnetoelastic effect yields a relatively higher current due to MEG's lower impedance. To substantiate this, we conducted experiments detailed as follows: Fig. R11a illustrates that our device operates using two layers - the MC and MI layers. During fabrication, the MC layer undergoes magnetization via a pulse magnetizer to induce magnetic flux changes. Initial tests with the standard fabrication process, using a shaker at 5 Hz (Fig. R11b), revealed the device generated a clear signal with an amplitude of 11 μA . **In a subsequent test, we employed an identical structure but refrained from magnetizing the MC layer, ensuring no magnetic flux density alterations.** Any resulting signal would solely be attributed to potential TENG effects. As presented in Fig. R11c, the absence of magnetization in the MC layer resulted in a negligible signal. A closer examination in Fig. R11d reveals only a noise-like signal with an amplitude of approximately 0.05 μA , effectively ruling out TENG's role in the device's current generation.

Fig. R11 | Output of MEGs with and without magnetization of the MC layer. **a**, Structure illustration of the MEG. **b**, Output of the device with the magnetization of the MC layer. **c**, Output of the device without the magnetization of the MC layer. **d**, Zoom in on Figure R10c.

We have also added a Supplementary Note 3 regarding this issue and have made corresponding clarifications in the main text as follows:

“To prove that there are no triboelectric components in the electric output of the skin-interfaced MEG devices, we identified several key factors necessary for the electrical conversion of TENG: 1. A contact-separation gap must exist between the MEG device and human skin. 2. A single electrode TENG must be grounded to facilitate charge exchange with the ground. Furthermore, the current output of a triboelectric nanogenerator (TENG) typically stays at the μA level, stemming from TENG's capacitive operating mechanism. However, our MEG typically produces a much higher current signal at the mA level. We delve into this comparison further and present detailed experimental results below:

1. Difference in structure design and operation mode. A typical MEG holds an all-in-one body design, there is no relative movement among layers. Furthermore, an adhesive layer was applied to prevent any potential sliding between the MEG device and the skin. As depicted in Fig. S39a, our MEG device is securely affixed to the skin with an adhesive layer, which prevents the relative sliding/movement between the MEG device and the skin. The sensing component consists of the MC layer and the MI layer, both of which are fully embedded in the PDMS matrix to form an all-in-one body design.

Fig. S39 | Device structure comparison between MEGs and TENGs. a, illustration of device structure of a MEG. **b**, illustration of device structure of a TENG with single electrode mode.

2. **Difference in electrode numbers and connection to loading resistance.** The MEG has two electrode outlets that were connected to the two ends of a loading resistance, while TENG only has one electrode outlet, going through the loading resistance, and then connected to the ground. The electrode connection in the circuit makes our device unable to satisfy the structure of a single-electrode TENG, as illustrated in Fig. S39b. For the single-electrode mode TENG, one circuit end is linked to an electrode that continuously interacts with another material possessing a different electronegativity, while the other end is grounded. This facilitates electron flow to the ground, producing a current. In our design, both electrode ends connect to a current meter or a loading resistance, making it implausible to replicate the single electrode TENG mode.
3. **Order of magnitude difference in electrical output.** The typical current output of a MEG is at the mA level, while the TENG is at the μA level. The typical voltage output of a MEG is at the 100-mV level, while the TENG is at the 100 V level. The typical inner impedance of a MEG is at the $10\ \Omega$ level, while that of a TENG is at the $\text{M}\Omega$ level. In a word, the electric output of TENGs is characteristically high voltage and low current due to their high internal impedance. Conversely, the magnetoelastic effect yields a relatively higher current due to the MEGs' lower internal impedance.

Fig. S40 | Output of MEG with and without magnetization of the MC layer. **a**, Structure illustration of the MEG. **b**, Output of the device with the magnetization of the MC layer. **c**, Output of the device without the magnetization of the MC layer. **d**, Zoom in on Figure R10c.

In addition, to substantiate this, we conducted experiments detailed as follows: Fig. S40a illustrates that our device operates using two layers - the MC and MI layers. During fabrication, the MC layer undergoes magnetization *via* a pulse magnetizer to induce magnetic flux changes. Initial tests with the standard fabrication process, using a shaker at 5 Hz (Fig. S40b), revealed the device generated a clear signal with an amplitude of 11 μA . In a subsequent test, we employed an identical structure but refrained from magnetizing the MC layer, ensuring no magnetic flux density alterations. Any resulting signal would solely be attributed to potential triboelectric effects. As presented in Fig. S40c, the absence of magnetization in the MC layer resulted in zero electric signal generation. A closer examination in Fig. S40d reveals only a noise signal with an amplitude of approximately 0.05 μA , effectively ruling out the TENG's role in the device's current generation.”

19. The conclusion section should be enriched with the most important quantitative results of the work, in terms of sensitivity, SNR, response time.

Response:

We appreciate the reviewer's insightful comment regarding the conclusion part. We have revised the part and included quantitative results of the work. Changes in the main text has been made as follows:

“In this work we have developed a wearable sensing-actuation system for assisted speaking without the need of vocal folds based on magnetoelastic effect in soft polymers. The device could translate the laryngeal muscle movement into voice signals, enabling communication without using the vocal fold during voice disorders' recovery process. We have tested and confirmed several attractive features of the device, including light weight

of 7.2g, high stretchability of a maximum strain of 164%, skin-like modulus of 7.83×10^5 Pa, isotropy, high SNR of 17.5, quick response time of 40 μ s, excellent sound producing quality, and water resistance. In addition, the device has been proven to detect unique, distinguishable signals of each syllabus from the laryngeal muscle movement without losing any essential waveform characteristics for downstream analysis. With the assistance of machine learning algorithm, the device can classify the semantic content of the movement signal and select the corresponding voice signal for outputting through the actuation component. Our device offers a practical solution for patients with voice disorders to communicate.”

20. How the MC layer is processed with the laser cutter? The authors should show some optical micrographs of the kirigami cuttings in order to give an idea of the sharpness of the cut edges.

Response:

We are very much thankful to the reviewer for emphasizing the importance of detailing the fabrication process of the MC layer and the clarity of the kirigami cut edges. The MC layer's fabrication involves a series of meticulous steps:

1. **Preparation of Magnetic Paste:** Initially, neodymium-iron-boron (NdFeB, Magnequench) magnetic powder is uniformly blended with the polydimethylsiloxane substrate (PDMS, Sylgard 184) at a mixing ratio of 15:1. The weight proportion between the magnetic powder and PDMS is maintained at 4:1.
2. **Molding and Curing:** The freshly prepared magnetic paste is then poured into a 3D-printed mold made of polylactic acid (PLA) with dimensions of 30 x 30 x 1 mm (length x width x height). This assembly is subsequently transferred to an oven, where it is set at 70°C for a duration exceeding 4 hours to ensure thorough curing.
3. **Magnetization:** After the curing process, the solidified MC layer is carefully extracted from the mold. It is then subjected to magnetization using an impulse magnetizer (IM-10-30, ASC Scientific). The magnetization is induced at an angle of 45° to the magnetization direction, utilizing an impulse voltage of 350 V.
4. **Laser Cutting:** The magnetized MC membrane is then positioned in a laser cutter (ULTRA R5000, Universal Laser System). The desired kirigami pattern is designed using AutoCAD software and subsequently uploaded to the laser cutter. To ensure precision and depth, the laser cutter is programmed to repeatedly trace the same pattern without repositioning the MC membrane. This iterative process ensures that the cuts progressively deepen until they fully penetrate the membrane, culminating in the desired kirigami structure.
5. **Optical Micrographs:** In response to the reviewer's suggestion, we have captured optical micrographs of the kirigami cuttings to provide a clear visualization of the sharpness and precision of the cut edges. These images can be found in **Figure R12**.

We hope this detailed description offers a clearer understanding of our fabrication process. Necessary modifications have also been incorporated into the main manuscript for clarity and completeness.

“The cured MC layer was then removed from the mold and magnetized by an impulse magnetizer (IM-10-30, ASC Scientific) with an induced angle of 45° to the magnetization direction at an impulse voltage of 350 V. The magnetized MC membrane is then positioned in a laser cutter (ULTRA R5000, Universal Laser System). The desired kirigami pattern is designed using AutoCAD software and subsequently uploaded to the laser cutter. To ensure precision and depth, the laser cutter is programmed to repeatedly trace the same pattern without repositioning the MC membrane. This iterative process ensures that the cuts progressively deepen until they fully penetrate the membrane, culminating in the desired kirigami structure.”

Fig. R12 | Optical micrograph of the kirigami structure on the MC layer, scale bar, 1mm.

In summary, we would like to take this opportunity to express our profound gratitude to the reviewers for their meticulous and insightful comments on our manuscript. Their keen observations and constructive feedback have been instrumental in highlighting the pivotal aspects of our research. Furthermore, their suggestions on the presentation and expression of our data have not only enhanced the clarity and robustness of our paper but also expedited its readiness for publication. The depth and precision of their reviews have undoubtedly elevated the quality of our work, and for that, we are truly thankful. In summary, their invaluable input has been pivotal in refining our manuscript, and we deeply appreciate the time and effort they dedicated to this process.

Reviewer #2:

This manuscript presents an interesting use of magnetoelastic materials for sensing vibrations, and the application to generate voice for speech impaired people. There are scientific merits that may benefit certain sensor and medical communities. However, there are flaws in data collection, presentation and experimental procedures that render this manuscript unsuitable to be published at its current stage. The strengths of this work as well as the issues are summarized below.

Response:

We are profoundly thankful to the reviewer for their recognition of our work. We genuinely appreciate their words of encouragement and valuable feedback, which encompassed phrases like “*Interesting*” and “*There are scientific merits that may benefit certain sensor and medical communities*”. Such positive appraisal and recommendation of the manuscript acceptance after addressing his/her questions, serves as a tremendous source of motivation, fueling our determination to consistently pursue excellence in our endeavors.

1. Although this paper was based on previous published work, there is a clear novelty in its kirigami design of the magnetoelastic coupling layer that is interesting and effective in increasing performance.

Response:

We sincerely appreciate the reviewer's recognition of the kirigami design implemented in our study, especially the acknowledgment of its "clear novelty" and its effectiveness in enhancing performance.

3. This technique, which uses magnetoelastic materials to detect vibrations, contains some technical innovations. However, practically, this approach may not be as impactful since piezoelectric systems can theoretically have similar performance but with a simpler design.

Response:

We deeply appreciate the reviewer's insightful feedback and recognition of the technical innovations in our work. While we acknowledge the potential of piezoelectric systems, we believe our magnetoelastic approach offers distinct advantages that enhance its practicality and impact. To further elucidate:

1. **Combing sensing and actuation in one system:** Our device includes two components, sensing and actuation. The magnetoelastic property enables two components in one system with very low driving voltage of 1.95V. While there are alternative materials that can be used for sensing, they may not offer comparable performance in actuation as magnetoelastic materials. For example, piezoelectric materials are commonly used for mechanical-electrical energy conversion. However, their driving voltage for actuation tasks, such as in ultrasound probes, is relatively high. This could raise safety concerns when integrated into wearable electronics.
2. **Simplified Design for Scalable Manufacturing:** Our device is designed with a focus on simplicity, consisting of just two primary layers: the MC and MI. This streamlined design is highly conducive to large-scale manufacturing. The MC layer's fabrication leverages established technologies used in producing silicon polymer products, offering flexibility in shape and material selection. The MI layer benefits from mature coil processing technology, a staple in the electronics industry. This combination not only ensures a broad material selection but also cost-effective manufacturing. For piezoelectric materials, the realization of soft and stretchable includes adopting materials of PVDF, which requires stretchable electrode to work; or using intricate array design as in reference [1,2]. Both requirements very sophisticated fabrication techniques and are difficult to realize for scalable manufacturing.
3. **Waterproof Advantage:** A distinguishing feature of our device is its inherent waterproof nature. In wearable applications, this is crucial as it effectively counters the challenges posed by perspiration and environmental humidity. This ensures the device's consistent performance across diverse scenarios, including sports wearables, underwater activities, and field operations. For piezoelectrical materials, the encapsulation layer for the water-proofness hinders the detection of minor biomechanical motions, especially in our case the minor movement of laryngeal muscles.
4. **Material Limitations of Piezoelectric Systems:** Piezoelectric materials have indeed made significant strides in wearable sensors, energy harvesting, and implantable bioelectronics. Their role in ultrasound-based biomedical applications is particularly noteworthy. However, these materials come with inherent challenges, such as high modulus, concerns about biocompatibility, and material degradation. These limitations can restrict material choices and some applications. In contrast, our magnetoelastic approach, as a novel self-powered mechanical-electrical energy converter, overcomes these challenges. It not only showcases potential in various wearable bioelectronics but also offers an alternative design choice for wearable devices.

In conclusion, while we recognize the capabilities of piezoelectric systems, we believe our magnetoelastic approach brings forth unique advantages that enhance its practicality and broaden its application spectrum.

Reference :

1. Lin, Muyang, et al. "A fully integrated wearable ultrasound system to monitor deep tissues in moving subjects." *Nature Biotechnology* (2023): 1-10.
2. Xu, Yuchen, et al. "In-ear integrated sensor array for the continuous monitoring of brain activity and of lactate in sweat." *Nature Biomedical Engineering* (2023): 1-14.
3. The authors claimed that the sensor is self-powered. It is true that although the signal is wirelessly generated by induction current, it still needs a powered device to measure the current. Thus, the wearable system will still need to be powered and the self-powered claim may be inaccurate.

Response:

We appreciate the reviewer's astute observation regarding the term "self-powered" and its application in our manuscript. In the context of our study, and as supported by prior research on "self-powered" sensors utilizing triboelectric and other mechanisms, the term "self-powered" typically denotes devices that "harvest energy from the body and its ambient environment, encompassing sources such as biomechanical, solar, thermal, and biochemical energy" for the primary sensing function¹⁻⁵. These sensors intrinsically generate their signals without external energy input but do require external power for subsequent data processing and analysis, as highlighted in the context of "real-time data transmission, mobile data processing, and smart power utilization"².

In our work, when we describe our sensor as a "self-powered sensor," we emphasize that the primary signal generation, derived from the biomechanical vibrations of the laryngeal muscles, is autonomous and does not require an external power source. The MC layer captures this biomechanical movement, converting it into an electrical signal through the magnetoelastic effect and electromagnetic induction. This process is fundamentally an electromechanical energy exchange, leading to current generation. Contrary to certain sensing modalities, like resistive sensors that demand an external circuit for signal acquisition, our sensing component inherently produces its signal. However, it's crucial to recognize that subsequent stages of signal processing and filtering necessitate external power management circuits.

To address this distinction more transparently, we have revised the main text as follows:

"This conversion represents an electromechanical energy exchange and is inherently a current-generating process. Unlike some sensing mechanisms, such as resistive sensors that necessitate an external circuit for signal detection, our sensing component operates in a self-powered manner. While additional power management circuits are essential for processing and filtering the signal, the initial sensing phase is autonomous and does not rely on an external energy supply, thus realizing self-powered biomechanical signal collection."

References:

1. Xu, S. et al. Self-powered nanowire devices. *Nature Nanotech.* **5**, 366–373 (2010).
2. Zheng, Q., Tang, Q., Wang, Z. L. & Li, Z. Self-powered cardiovascular electronic devices and systems. *Nat. Rev. Cardiol.* **18**, 7–21 (2021).
3. Luo, J. et al. Flexible and durable wood-based triboelectric nanogenerators for self-powered sensing in athletic big data analytics. *Nat. Commun.* **10**, 5147 (2019).
4. Zhang, F., Zang, Y., Huang, D., Di, C. & Zhu, D. Flexible and self-powered temperature–pressure dual-parameter sensors using microstructure-frame-supported organic thermoelectric materials. *Nat. Commun.* **6**, 8356 (2015).
5. Sengupta, S. et al. Self-powered enzyme micropumps. *Nat. Chem.* **6**, 415–422 (2014).
6. Most data do not include repetitions or multiple samples. In typical sensor/device validation experiments, data are typically collected multiple times with multiple samples to ensure the measurements are consistent across different samples and are repeatable. While it makes sense that illustrative plots such as Fig 3e presents only one measurement from a single sample, performance-based plots such as Fig 2b and 2f should have data points that are the average of multiple samples (and multiple measurements per sample) with error bars indicating the standard deviation among the measurements.

Response:

We are deeply grateful to the reviewer for emphasizing the importance of data repetitions and multiple samples in our experiments. Acknowledging this crucial feedback, we have revisited our experiments and incorporated error bars in all relevant figures throughout the manuscript to reflect the standard deviation among measurements. Below, we present the updated Figure 2 as a representative example of these revisions. We believe these modifications enhance the rigor and clarity of our data presentation.

Fig. R13. Revised Figure 2 | Performance characterization of the wearable sensing-actuation system. a, Performance comparison of different flexible throat sensor in terms of the Young’s modulus, stretchability, under water sound pressure level, temperature rise, driving voltage, and working frequency range. **b,** Pressure-Sensitivity response of the device at varied degrees of stretching under different amplification levels. **c,** Response time and signal-to-noise ratio of the device. **d,** Variation of sound pressure level with distance from the device at different amplification levels. **e,** Sound pressure level of the device with resonance point highlighted in human hearing frequency range compared to SPL normal human speaking threshold. **f,** The right shift of first resonance point towards high frequency with regards to increasing strains. **g,** Relationship Between Kirigami Structure Parameters and Actuating (First resonance point and sound pressure level) /Sensing Properties (Response time and Signal to noise ratio). **Waveform (h) and spectrum (i) comparison of commercial loudspeaker (Red) and the device (Yellow) sound output at 900Hz and maximum strain (164%).**

7. The methodology used for validating the sensor is mostly sound. However, many details are missing thus making reproducing the sensor by other labs impossible. Some missing or inconsistent details such as the following should be included.

- “The entire system is small, thin (~ 1.35 ml)...” It is unclear if 1.35 ml is describing the volume of the sensor. If so, mm³ may be a better unit to use.

Response:

We are grateful to the reviewer for pointing out the unit discrepancy in describing the sensor's volume. Indeed, when referencing the volume of the sensor, 1.35 mm^3 is the appropriate unit. We have updated the manuscript to reflect this correction. Thank you for bringing this to our attention.

“The entire system is small, thin ($\sim 1.35 \text{ cm}^3$, with a width and length of $\sim 30 \text{ mm}$ and a thickness of $\sim 1.5 \text{ mm}$), and lightweight ($\sim 7.2268 \text{ g}$) (see Fig. S2 and Supplementary Table S1).”

8. In the description that the authors used magnetization impulse field to magnetize the substrate, please provide the direction and magnitude of the field.

Response:

We are grateful to the reviewer for highlighting the need for clarity regarding the magnetization process. To address this, the MC layer was magnetized using an impulse magnetizer (IM-10-30, ASC Scientific). Specifically, an induced angle of 45° relative to the magnetization direction was applied with an impulse voltage of 350 V, as illustrated in **Figure R14a**. This angle was chosen to ensure signal generation in both vertical and horizontal directions in-plane. Figure R14b provides a top-down view of the magnetizing process, showing the device positioned in the magnetization chamber such that its diagonal aligns parallel to the magnetization direction. We have incorporated these details into the main text for clarity.

“The cured MC layer was then removed from the mold and magnetized by an impulse magnetizer (IM-10-30, ASC Scientific) with an induced angle of 45° to the magnetization direction at an impulse voltage of 350 V.”

Fig. R14 | Magnetization detail illustration of the MC layer. a, the illustration of the magnetization process. **b**, vertical view of the magnetization process of the device in the magnetization chamber.

9. What is the spacing between the loops of the copper coil?

Response:

We deeply appreciate the reviewer for their constructive and professional comment regarding the spacing between loops of the copper coil. We have measured and calculated the space with summation method to be $22.3 \pm 2.14 \mu\text{m}$.

We have also included this information in the main text :

“A serpentine-shaped 3D printed mold is used to twine a copper coil with a spacing of $22.3 \pm 2.14 \mu\text{m}$ ”

11. Under Electrical Performance Measurement, the current signal of the device was measured with a preamp. Please provide the gain and filter setting used to perform the measurement.

Response:

We are thankful to the reviewer for underscoring the necessity of detailing the settings used during the current measurement. To ensure clarity and transparency, here are the specific settings employed with the Stanford low-noise current preamplifier (model SR570) during our experiments:

Gain Mode: We selected the "LOW NOISE" mode to ensure the most accurate and noise-free measurements.

Sensitivity: This was adjusted to " $2 \times 100 \mu\text{A/V}$ ", which allowed us to capture even minute variations in the current.

Filter Frequency: We employed a "Lowpass 6 dB" filter set at "100 Hz". This setting was chosen to filter out any high-frequency noise that could interfere with our measurements.

Input Offset: This was set to "NEG" with a value of " $1 \times 10 \mu\text{A}$ " to account for any inherent offset in the preamplifier.

We believe that these settings provided us with the most accurate and consistent current measurements for our device. In light of the reviewer's feedback, we have incorporated these details into the 'Electrical Performance Measurement' section of the main manuscript to ensure clarity for all readers.

"Electrical Performance Measurement. The current signal of the device is measured by a current Stanford low-noise current preamplifier (model SR570) with the following parameters:

Gain Mode: We selected the "LOW NOISE" mode to ensure the most accurate and noise-free measurements.

Sensitivity: This was adjusted to " $2 \times 100 \mu\text{A/V}$ ", which allowed us to capture even minute variations in the current.

Filter Frequency: We employed a "Lowpass 6 dB" filter set at "100 Hz". This setting was chosen to filter out any high-frequency noise that could interfere with our measurements.

Input Offset: This was set to "NEG" with a value of " $1 \times 10 \mu\text{A}$ " to account for any inherent offset in the preamplifier."

12. please provide more information (physical and magnetic properties) of the neodymium-iron-boron (NdFeB), such as the size of the particles, the magnetization and coercivity, etc.

Response:

We appreciate the reviewer's meticulous attention to the details of our manuscript. Regarding the neodymium-iron-boron (NdFeB, Magnequench) magnetic powder, we provide the following specifications:

Particle Size (D50): 5 μm

Residual Induction (Br): 898-908 mT, 8.98-9.08 kG

Energy Product (BH) max: 120-128 kJ/m³, 15.0-16.0 MGOe

Intrinsic Coercivity (Hci): 700-740 kA/m, 8.8-9.3 kOe

Magnetizing Field to >95% Saturation (Min.): $H_s \geq 1600 \text{ kA/m}$, $\geq 20.0 \text{ kOe}$

Coercive Force (Hc): 515 kA/m, 6.5 kOe

We have incorporated this detailed information into the main text to ensure clarity and provide a comprehensive understanding of the material properties:

“The neodymium-iron-boron (NdFeB, Magnequench) magnetic powder with the following properties is used in the study: Particle Size (D50), 5 μ m; Residual Induction (Br): 898-908 mT, 8.98-9.08 kG; Energy Product (BH) max: 120-128 kJ/m³, 15.0-16.0 MGOe; Intrinsic Coercivity (Hci): 700-740 kA/m, 8.8-9.3 kOe; Magnetizing Field to >95% Saturation (Min.): $H_s \geq 1600$ kA/m, ≥ 20.0 kOe; Coercive Force (Hc): 515 kA/m, 6.5 kOe. The magnetic powder is evenly mixed with polydimethylsiloxane substrate (PDMS, Sylgard 184) at a mixing ratio of 15:1.”

13. Regarding the fabrication of the magnetic coupling layer, the authors stated that NdFeB magnetic powder vs PDMS mixing ratio of 15:1, but the weight ratio is 4:1? Please elaborate why there are two different mixing ratios.

Response:

We are grateful to the reviewer for pointing out the ambiguity in our description of the mixing ratios. To clarify:

1. The ratio of 15:1 pertains to the mixing of the PDMS elastomer base and its curing agent. Specifically, for every 15 parts of the elastomer base, 1 part of the curing agent is added.
2. The 4:1 weight ratio refers to the proportion of the NdFeB magnetic powder to the combined PDMS mixture (which includes both the elastomer base and the curing agent). In essence, for every 4 parts of magnetic powder, 1 part of the fully mixed PDMS (elastomer base and curing agent) is used.

We regret the oversight and have made the necessary clarifications in the manuscript to ensure there's no confusion regarding the fabrication process.

“The magnetic powder is evenly mixed with polydimethylsiloxane substrate (PDMS, Sylgard 184). The PDMS is fabricated with its elastomer base and its curing agent mixed at a ratio of 15:1. Subsequently, the weight ratio of the magnetic powder and mixed-PDMS is measured to be 4:1.”

14. Regarding the investigation of sensor performance with or without sweat, it is unclear what is the procedure to produce sweat in the test subjects, or how much sweat on skin was being tested.

Response:

We appreciate the reviewer's attention to the details of our experimental procedure. To evaluate the device's resilience and performance under sweat-like conditions, we employed artificial sweat surrogate (Biochemazone Inc., Artificial Sweat BZ320). The consistent composition of artificial sweat ensured uniformity across all tests. The procedure for sweat simulation and device testing was as follows:

1. Skin Preparation: Each participant's throat area was meticulously cleaned using an alcohol pad to eliminate any natural oils or residues. After this, the area was dried with a tissue pad to ensure the complete removal of residual alcohol.
2. Initial sweat Application: A calibrated spray bottle was utilized to evenly apply 0.5 ml of artificial sweat solution onto the cleaned skin area, simulating a layer of sweat.
3. Device Attachment: Post the artificial sweat application, the device was carefully affixed to the treated skin surface, ensuring optimal contact.
4. Secondary sweat Application: To further mimic sweat exposure, an additional 0.5 ml of artificial sweat solution was sprayed directly onto the device's surface.
5. Settling Period: Participants were then instructed to remain stationary for a duration of 5 minutes. This interval was crucial to assess any potential infiltration of the artificial sweat solution into the device.
6. Data Collection: Following the settling period, the device's performance metrics were recorded under the simulated sweat conditions.

We have incorporated these clarifications into the main text to provide a clearer understanding of our methodology.

“Sweat simulation test. To evaluate the device's resilience and performance under sweat-like conditions, we employed artificial sweat surrogate (Biochemazone Inc., Artificial Sweat BZ320). The consistent composition of

artificial sweat ensured uniformity across all tests. The procedure for sweat simulation and device testing was as follows:

1.Skin Preparation: Each participant's throat area was meticulously cleaned using an alcohol pad to eliminate any natural oils or residues. Subsequent to this, the area was dried with a tissue pad to ensure the complete removal of residual alcohol.

2.Initial sweat Application: A calibrated spray bottle was utilized to evenly apply 0.5 ml of artificial sweat solution onto the cleaned skin area, simulating a layer of sweat.

3.Device Attachment: Post the artificial sweat application, the device was carefully affixed to the treated skin surface, ensuring optimal contact.

4.Secondary sweat Application: To further mimic sweat exposure, an additional 0.5 ml of artificial sweat solution was sprayed directly onto the device's surface.

5.Settling Period: Participants were then instructed to remain stationary for a duration of 5 minutes. This interval was crucial to assess any potential infiltration of the artificial sweat solution into the device.

6.Data Collection: Following the settling period, the device's performance metrics were recorded under the simulated sweat conditions."

15. Validation experiment for the machine learning algorithm is still under very controlled situations, thus it is unclear if the sensor/AI can actually be used practically. For example, there is a lack of description on what sentences were used in the machine learning experiment, thus the accuracy of the algorithm may only true under very narrow conditions.

Response:

We appreciate the reviewer for insightful feedback regarding the validation of our machine learning algorithm. We understand the importance of clarity in describing the conditions under which our algorithm was tested to ensure its practical applicability. Our study employed a machine-learning algorithm to interpret laryngeal muscle movement data and select an appropriate voice signal for output. The process involves two main steps: training and classifying a set of sentences. The electrical signals of these sentences were transformed into matrices for feature extraction using principal component analysis (PCA). This helped in removing redundancy and preparing the signal for classification. The classification was achieved using a multi-class support vector classification (SVC) with a "one vs rest" decision function. After training, the classifier was used to identify real-time laryngeal muscle movement signals. A demonstration was conducted with five sentences, with each participant repeating each sentence 100 times. The classifier achieved an overall prediction accuracy of 94.68% across different participants. The device's sound performance was consistent across various conditions, including the presence of perspiration and different conversation angles.

Regarding the choice of the SVM method, we opted for this approach because the data used to train the model is one-dimensional waveform data. Simpler algorithms, like the one we employed, tend to perform better on datasets that aren't information dense. To address your concern about the sentences used in the machine learning experiment: we have provided a description of the specific sentences used in our study (S1:" Hi Rachel, how you doing today?", S2:" Hope your experiments are going well!", S3:" Merry Christmas!", S4:" I love you!", S5:" I don't trust you."). We included sentences with varying syntactical structures, lengths, tones, and complexities, such as "Hi Rachel, how are you doing today?" and "I don't trust you." This diverse selection ensures our system's practical feasibility across a broad range of scenarios.

Corresponding changes has also been made in the main text and Figure 4 pointing out the sentence we adopted in this study:

“A brief demonstration was made with five sentences that we had selected for training the algorithm (S1:” Hi Rachel, how you doing today?”, S2:” Hope your experiments are going well!””, S3:” Merry Christmas!””, S4:” I love you!””, S5:” I don’t trust you.”).”

Fig. R15. Revised Figure 4 | Machine-learning-assisted wearable speaking without vocal folds. a, Flow chart of the machine-learning-assisted wearable sensing-actuation system. **b**, Illustration depicting the process of data segmentation and Principal Components Analysis (PCA) applied to the muscle movement signal captured by the sensor. **c**, Optimizing process of data classification after PCA with Support Vector Machine (SVM) algorithm. **d**, Contour plot of the classification result with SVM, class “1” indicating 100% possibility of the target sentence, dotted lines are the possibility boundaries between the target sentence and the others. **e**, Bar chart exhibiting 7 participants’ accuracy of both validation set and testing set. **f**, Confusion matrix of the 8th participant’s validation set with an overall accuracy of 98%. **g**, Confusion matrix of the 8th participant’s testing set with an overall accuracy of 96.5%. **h**, Demonstration of the machine-learning-assisted wearable sensing-actuation system in assisted speaking. The left panel shows the muscle movement signal captured by the sensor as the participant pronounces the sentence voicelessly, while the right panel shows the corresponding output waveform produced by the system’s actuation component. **i**, The SPL and temperature trends over time while the device is worn by participants; no notable temperature increase or SPL decrease was seen for up to 40 minutes. **j**, The device’s SPL as it outputs participant-specific sound signals, both with and without sweat presence. **k**, The device’s SPL across various conversation angles while donned by the participant.

16. This manuscript can use more proof readings. There are obvious grammatical errors.

Response:

We sincerely apologize for the oversight and any confusion caused by the grammatical errors present in the manuscript. We value the importance of clear and precise communication in scientific writing. To ensure the quality of our manuscript, we had indeed conducted multiple rounds of proofreading and utilized grammar-checking tools. However, it seems that some errors still escaped our attention. We appreciate the reviewer's patience and feedback on this matter. We have taken additional measures, including seeking external professional editing services, to ensure that the revised manuscript is free from grammatical errors and is presented in the best possible manner. We are committed to upholding the highest standards of clarity and accuracy in our work.

In summary, we greatly appreciate the positive and constructive comments from the two reviewers on our manuscript, which guided us to make extensive investigations in detail to elaborate our points as well as to fully justify the significance of this work in the past 2 months. The insightful suggestions and corresponding revision greatly consolidated our work to benefit future readers with a better understanding.

REVIEWERS' COMMENTS

Reviewer #1 (Remarks to the Author):

The authors have addressed thoroughly and with lots of details all the points raised by the reviewer, who is then pleased to recommend the manuscript for publication.

Just two small notes:

- Some references for the description of PVDF would be beneficial.
- Regarding the comparison between MEG and TENG, the authors should include the suggested reference since it was useful to make the new experiments.

Reviewer #2 (Remarks to the Author):

The authors have addressed all concerns and suggestions from my previous review and have made satisfactory revisions, as well as provided detailed description of the changes. There are no further major concerns.

Minor suggestion:

A better color usage in plots may prevent misunderstanding. For example, in Figure 2g, the SNR error band, which is colored light grey, is barely visible (at least in my PDF version). Same with some texts in Figure 4j. I suggest using darker color or shading to avoid losing figure definition due to file format conversion.

Response to Reviewers' Comments (manuscript NCOMMS-23-37367A)

We extend our deepest appreciation to the reviewers for their thorough and insightful feedback during the review process. Their detailed comments and constructive suggestions have been invaluable in refining and strengthening our manuscript. We are particularly grateful for the guidance provided in enhancing the experimental details and improving the presentation of our figures, which has significantly elevated the quality and clarity of our work. The opportunity to incorporate such thoughtful recommendations has not only enriched our manuscript but also contributed to our professional growth as researchers. We believe these final revisions have addressed the concerns raised and have further solidified the contribution of our study to the field.

Reviewer #1:

The authors have addressed thoroughly and with lots of details all the points raised by the reviewer, who is then pleased to recommend the manuscript for publication.

Response:

We sincerely appreciate Reviewer #1's positive recognition of our efforts to thoroughly address the points raised in their review. It is heartening to hear that our detailed responses and revisions have met with your approval and have led to a recommendation for publication. We are deeply grateful for your acknowledgment of the depth and detail of our work in advanced bioelectronics and materials engineering. Your encouraging words and constructive feedback have been instrumental in enhancing the quality of our manuscript.

1. Some references for the description of PVDF would be beneficial.

Response:

We are grateful to the reviewer for their constructive feedback on the inclusion of reference of PVDF. We recognize the importance of providing references for the description of properties of PVDF and have added references as follows:

“Existing research on medical devices using flexible loudspeakers and wearable throat sensors, made from materials like polyvinylidene fluoride (PVDF)³⁵⁻³⁸, gold nanowires³⁹, or graphene^{40,41}, has shown potential for aiding communication during recovery from vocal fold disorders. PVDF emerges as a pristine thermoplastic fluoropolymer, notable for its exceptional non-reactivity⁴². A distinguishing feature of PVDF is its piezoelectric property, adeptly converting mechanical oscillations into precise voltage signals^{43,44}.”

Reference:

42. Ribeiro, C. et al. Electroactive poly(vinylidene fluoride)-based structures for advanced applications. *Nat. Protoc.* 13, 681–704 (2018).
43. Li, M. et al. Revisiting the δ -phase of poly(vinylidene fluoride) for solution-processed ferroelectric thin films. *Nat. Mater.* 12, 433–438 (2013).
44. Katsouras, I. et al. The negative piezoelectric effect of the ferroelectric polymer poly(vinylidene fluoride). *Nat. Mater.* 15, 78–84 (2016).

2. Regarding the comparison between MEG and TENG, the authors should include the suggested reference since it was useful to make the new experiments.

Response:

We would like to express our appreciation to the reviewer for raising his/her question by adding the corresponding reference regarding TENG to the article. We have added the reference and have also made corresponding changes in the main text:

“We have also examined the electric output of the device is not due to the triboelectricity in Supplementary Note 3⁶³.”

Reference:

63. Mariello, M., Fachechi, L., Guido, F. & De Vittorio, M. Conformal, Ultra-thin Skin-Contact-Actuated Hybrid Piezo/Triboelectric Wearable Sensor Based on AlN and Parylene-Encapsulated Elastomeric Blend. *Adv. Funct. Mater.* 31, 2101047 (2021).

Reviewer #2:

The authors have addressed all concerns and suggestions from my previous review and have made satisfactory revisions, as well as provided detailed description of the changes. There are no further major concerns.

We sincerely appreciate Reviewer #2's thoughtful and constructive feedback on our manuscript. We are grateful for your recognition of the effort and diligence we applied in addressing the comments and suggestions from your previous review. Your positive assessment of the revisions and detailed descriptions we provided is immensely encouraging.

1. A better color usage in plots may prevent misunderstanding. For example, in Figure 2g, the SNR error band, which is colored light grey, is barely visible (at least in my PDF version). Same with some texts in Figure 4j. I suggest using darker color or shading to avoid losing figure definition due to file format conversion.

Response:

We appreciate the reviewer's suggestions and recognize color usages in the figure. And we have adopted a darker color in the figures to avoid misread as the reviewer suggests as follows. Those changes have also been made in the main text:

Fig. R1. Revised Fig. 2 | Performance characterization of the wearable sensing-actuation system. **a**, Performance comparison of different flexible throat sensors in terms of the Young's modulus, stretchability, underwater sound pressure level, temperature rise, driving voltage, and working frequency range. **b**, Pressure-Sensitivity response of the device at varied degrees of stretching under different amplification levels. **c**, Response time and signal-to-noise ratio of the device. **d**, Variation of sound pressure level with distance from the device at different amplification levels. **e**, Sound pressure level of the device with resonance point highlighted in human hearing frequency range compared to SPL normal human speaking threshold. **f**, The right shift of first resonance point towards high frequency with regards to increasing strains. **g**, Relationship Between Kirigami Structure Parameters and Actuating (First resonance point and sound pressure level) /Sensing Properties (Response time and Signal to noise ratio). Waveform (**h**) and spectrum (**i**) comparison of commercial loudspeaker (Red) and the device (Yellow) sound output at 900Hz and maximum strain (164%).

Fig. R2. Revised Figure 4 | Machine-learning-assisted wearable speaking without vocal folds. **a**, Flow chart of the machine-learning-assisted wearable sensing-actuation system. **b**, Illustration depicting the process of data segmentation and Principal Components Analysis (PCA) applied to the muscle movement signal captured by the sensor. **c**, Optimizing process of data classification after PCA with Support Vector Machine (SVM) algorithm. **d**, Contour plot of the classification result with SVM, class "1" indicating 100% possibility of the target sentence, dotted lines are the possibility boundaries between the target sentence and the others. **e**, Bar chart exhibiting 7 participants' accuracy of both validation set and testing set. **f**, Confusion matrix of the 8th participant's validation set with an overall accuracy of 98%. **g**, Confusion matrix of the 8th participant's testing set with an overall accuracy of 96.5%. **h**, Demonstration of the machine-learning-assisted wearable sensing-actuation system in assisted speaking. The left panel shows the muscle movement signal captured by the sensor as the participant pronounces the sentence voicelessly, while the right panel shows the corresponding output waveform produced by the system's actuation component. **i**, The SPL and temperature trends over time while the device is worn by participants; no notable temperature increase or SPL decrease was seen for up to 40 minutes. **j**, The device's SPL outputs participant-specific sound signals, both with and without sweat presence. **k**, The device's SPL across various conversation angles while done by the participant.

In conclusion, we extend our deepest gratitude to both reviewers for their constructive and insightful feedback on our manuscript. Your thorough evaluations prompted us to undertake further detailed investigations, enabling us to enhance the clarity of our arguments and more effectively demonstrate the significance of our research. The thoughtful suggestions provided by each reviewer played a pivotal role in refining our work, ensuring that it not only meets but exceeds the standards of academic rigor. We firmly believe that these revisions have significantly strengthened our manuscript, offering future readers a more comprehensive and profound understanding of the subject. We are thankful for the opportunity to improve our work through this collaborative and enriching review process.